# Random Sharpness-Aware Minimization

**Yong Liu[1], Siqi Mai[2], Minhao Cheng[3], Xiangning Chen[4], Cho-Jui Hsieh[4], Yang You[1]**
[1]Department of Computer Science, National University of Singapore
[2]HPC-AI Technology Inc. [3]Department of Computer Science and Engineering, HKUST
[4]Department of Computer Science, University of California, Los Angeles
{liuyong, youy}@comp.nus.edu.sg, minhaocheng@ust.hk,
{xiangning, chohsieh}@cs.ucla.edu,

## Abstract

Currently, Sharpness-Aware Minimization (SAM) is proposed to seek the parameters that lie in a flat region to improve the generalization when training neural networks. In particular, a minimax optimization objective is defined to find the maximum loss value centered on the weight, out of the purpose of simultaneously minimizing loss value and loss sharpness. For the sake of simplicity, SAM applies one-step gradient ascent to approximate the solution of the inner maximization. However, one-step gradient ascent may not be sufficient and multi-step gradient ascents will cause additional training costs. Based on this observation, we propose a novel random smoothing based SAM (R-SAM) algorithm. To be specific, R-SAM essentially smooths the loss landscape, based on which we are able to apply the one-step gradient ascent on the smoothed weights to improve the approximation of the inner maximization. Further, we evaluate our proposed R-SAM on CIFAR and ImageNet datasets. The experimental results illustrate that R-SAM can consistently improve the performance on ResNet and Vision Transformer (ViT) training.

## 1   Introduction

Deep learning has seen remarkable progress in a variety of areas, with advanced development in over-parameterized neural networks. However, over-parameterization usually leads to overfitting and suffers poor generalization for unseen data [31]. To this end, many studies have investigated the generalization of deep neural network to narrow down the gap [7, 17, 26, 29, 42]. In particular, they try to analyse the local minima problem and connect it with loss geometry [17, 27, 31]. Amongst them, Foret et al. propose SAM, with an aim to solve a minimax optimization problem. SAM initially finds the maximum of loss function in the region of radius $\rho$ centered on the $w$, regarding which to minimize the loss value. This will prompt the model to converge to a flat region, where the entire neighborhood tends to have uniformly low training loss. Inner maximization is a vital part of SAM procedure as it essentially describes the loss landscape and plays an important role in whether we can reduce the loss sharpness effectively. The solution for inner maximization is NP-hard, and for simplicity, SAM uses one-step gradient ascent directly to approximate the maximum loss. However, one-step gradient ascent may not be sufficient. In addition, multi-step gradient ascents are not efficient and will cause additional training costs. Therefore, how to improve the approximation of the inner maximization without additional computation cost is an important problem.

Based on our empirical analysis in Figure 1 and Theorem 1, we find that random initialization can smooth the loss landscape and obtain a more stable gradient direction. Noted that a more stable gradient can make a good approximation to the true maximum value. Therefore, in this paper, we propose a novel random smoothing based sharpness-aware minimization algorithm (R-SAM). Our proposed R-SAM consists of two steps. First, we use a Gaussian noise to smooth the loss landscape and escape from the local sharp region to obtain a stable gradient for gradient ascent.

36th Conference on Neural Information Processing Systems (NeurIPS 2022).

Second, we perform the one-step gradient ascent on the smoothed weights to eventually enhance the approximation quality of inner maximization.

We summarize our contributions as follows:

- In this paper, we investigate the approximation quality of this one-step gradient ascent. Our evaluations show that unstable gradients might hurt the inner maximization of SAM and adding random smoothing can improve gradient stability.

- Based on our observation, we propose a simple but novel algorithm, R-SAM, which leverages a Gaussian noise in perturbation initialization and successfully enhance the inner maximization without introducing additional cost.

- We conduct empirical studies on ResNet and ViT. The experimental results demonstrate that R-SAM can help model generalization ability and yield more remarkable performance compared with SAM across a wide range of computer vision tasks (e.g., CIFAR-10/100, ImageNet).

## 2 Background

### 2.1 Sharpness-Aware Minimization

Many studies focus on analyzing the generalization of deep neural networks. For example, motivated by the local geometry of the energy landscape, Chaudhari et al. propose Entropy-SGD to construct a local-entropy-based objective function that favors well-generalizable solutions lying in large flat regions of the energy landscape, to address generalization problems in the training. Izmailov et al. illustrate the benefits of Stochastic Weight Averaging (SWA) procedure in solving generalization problems and well apply SWA to generate much flatter solutions than SGD.

Keskar et al. shows that deep neural network training is easy to converge to sharp local minima and causes a generalization gap. Therefore, how to make the model converge to a flat local minima and improve the generalization is an important problem. Foret et al. propose SAM to seek the parameters that lie in a flat region through optimizing loss value and loss sharpness. Then we will overview Sharpness-Aware Minimization (SAM).

Consider the training dataset as $\mathcal{S} = \{(x_i, y_i)\}_{i=1}^n$, where each sample $(x_i, y_i)$ follows the distribution $\mathcal{D}$. Define $f(x; \boldsymbol{w})$ as the neural network model with trainable parameter $\boldsymbol{w} \in \mathbb{R}^p$, with which the loss function w.r.t. an input $x_i$ can be given by $l(f(x_i; \boldsymbol{w}), y_i) \in R^+$, shortened to $l(x_i)$ for convenience. The empirical training loss is $L = \frac{1}{n} \sum_{i=1}^n l(f(x_i; \boldsymbol{w}), y_i)$. The objective of SAM [18] is to find the parameters with uniformly low loss value within the $\ell_p$ ball. The objective function is proposed as follows.

$$L^{SAM}(\boldsymbol{w}) = \max_{\|\boldsymbol{\delta}\|_p \leq \rho} L(\boldsymbol{w} + \boldsymbol{\delta}), \tag{1}$$

where $p \geq 0$ is defined as the radius of the $\ell_p$ ball. For simplicity, we ignore $p$ when using $\ell_2$-norm. As calculating the optimal solution of inner maximization is infeasible, SAM uses one-step gradient ascent to approximate it:

$$\hat{\boldsymbol{\delta}}(\boldsymbol{w}) = \rho \nabla_{\boldsymbol{w}} L(\boldsymbol{w}) / \|\nabla_{\boldsymbol{w}} L(\boldsymbol{w})\| \approx \arg \max_{\|\boldsymbol{\delta}\| \leq \rho} L(\boldsymbol{w} + \boldsymbol{\delta}). \tag{2}$$

Finally, SAM computes the gradient regarding the perturbed model $\boldsymbol{w} + \hat{\boldsymbol{\delta}}$ for the update:

$$\nabla_{\boldsymbol{w}} L^{SAM}(\boldsymbol{w}) \approx \nabla_{\boldsymbol{w}} L(\boldsymbol{w})|_{\boldsymbol{w} + \hat{\boldsymbol{\delta}}}. \tag{3}$$

Although SAM can achieve a great performance in traditional computer vision [8] and NLP tasks [3], there still exists several drawbacks yet to be resolved. To further improve its performance, Kwon et al. introduce the concept of adaptive sharpness of loss function to solve the scale-dependency problem. Zhuang et al. propose GSAM to seek the region with both small loss value and low sharpness based on a novel optimization objective. Zhou & Chen try to learn a dynamically reweighted perturbation for each batch to obtain a better approximation to per-instance perturbation. Noted that SAM needs to calculate two sequential gradients. To improve its efficiency, Du et al. propose ESAM, which use stochastic weight perturbation and sharpness-sensitive data selection to reduce the computation of vanilla SAM. Liu et al. propose LookSAM to reuse the projected gradient that target to the flat region

every $k$ steps to accelerate ViT training. SS-SAM is proposed to improve the efficiency by randomly selecting SGD optimization or SAM at each step [51].

## 2.2 Generalization

There is a venerable line of important works on the problem of generalization in deep learning[5, 12, 21, 28, 30]. For example, Hardt et al. focus on generating generalization bounds for models learned with stochastic gradient descent. Bisla et al. investigate the sharpness of loss landscape and proposed systematic mechanisms underlying the generalization abilities of deep learning models. Jin et al. analyse the importance of perturbed gradient descent in efficiently escaping saddle points. Further, Jin et al. investigate the use of a Hamiltonian function and propose a new framework to track the long-term behavior of gradient-based optimization algorithms. In addition, Dinh et al. argue that the concepts of flatness might be problematic and can not be directly applied to explain generalization, which requires rethinking what flatness actually means.

Adversarial training has achieved a great success in improving the robustness of model [13, 20, 39, 40, 45]. Our approach was also inspired by recent works in random smoothing in adversarial training[1, 9, 44, 47, 50]. Wong et al. demonstrate that applying random initialization in FGSM adversarial training can in fact be just as effective as the more costly PGD adversarial training. Further, Andriushchenko & Flammarion study the questions of when and why FGSM adversarial training works, and how to enhance the solution of the inner maximization problem. Zhang et al. propose a novel bi-level optimization-based fast adversarial training framework, FAST-BAT to address the issue of overfitting and achieve better performance in generalization ability.

# 3 Proposed Method

## 3.1 Empirical Observation on Approximation of Inner Maximization

As shown in Section 2.1, SAM defines a min-max optimization objective to find the maximum loss value $L^{SAM}(\boldsymbol{w})$ near the current weight $\boldsymbol{w}$ and then minimize the value $L^{SAM}(\boldsymbol{w})$. However, for the inner maximization problem, finding the optimal solution is NP-hard, which means that it's difficult to obtain the true maximum value. Therefore, SAM uses a one-step gradient ascent to approximate the optimal solution and obtain the maximum loss value. In this paper, we investigate the approximation quality of this one-step gradient ascent and propose a simple way to improve the inner maximization without introducing additional cost.

**Inner maximization of SAM may be hurt by unstable gradient.** The adversarial weight $\boldsymbol{w_{adv}}$ with the optimal solution of inner maximization can be defined as:

$$\boldsymbol{w_{adv}^*} = \boldsymbol{w} + \arg \max_{\|\boldsymbol{\delta}\|_p \leq \rho} L(\boldsymbol{w} + \boldsymbol{\delta}). \tag{4}$$

SAM uses a one-step gradient ascent to approximate the optimal adversarial weight:

$$\boldsymbol{w_{adv}^*} = \boldsymbol{w} + \arg \max_{\|\boldsymbol{\delta}\|_p \leq \rho} L(\boldsymbol{w} + \boldsymbol{\delta}) \approx \boldsymbol{w} + \rho \frac{\nabla_{\boldsymbol{w}} L(\boldsymbol{w})}{\|\nabla_{\boldsymbol{w}} L(\boldsymbol{w})\|}. \tag{5}$$

However, the loss landscape of neural network is usually sharp [31, 35]. That means the direction of $\boldsymbol{g}(\boldsymbol{w}) = \nabla_{\boldsymbol{w}} L(\boldsymbol{w})$ cannot represent the direction to maximize its neighbourhoods' loss value. However, a stable gradient direction in a continuous parameter space can narrow the approximation gap and help SAM obtain a better approximation of $\boldsymbol{w_{adv}^*}$. Therefore, we try to conduct the experiments in ResNet and WideResNet to analyse the smoothness of gradient $\boldsymbol{g}(\boldsymbol{w})$. As shown in Figure 1, we plot the cosine value between the $\boldsymbol{g}(\boldsymbol{w})$ and $\boldsymbol{g}(\boldsymbol{w} + \rho \frac{\boldsymbol{g}(\boldsymbol{w})}{\|\boldsymbol{g}(\boldsymbol{w})\|})$ to visualize the smoothness. From this figure, we can find that the cosine value decreases significantly with the $\rho$ value increasing (the red line). Intuitively, when the gradient is constant in a certain region (cosine similarity equals to 1), the one-step gradient ascent will get the exact maximizer; while when the gradient becomes unstable in a certain region, the one-step gradient ascent will lead to a poor solution. Therefore, how to smooth the loss landscape and make the gradient stable during the gradient ascent process is an important problem.

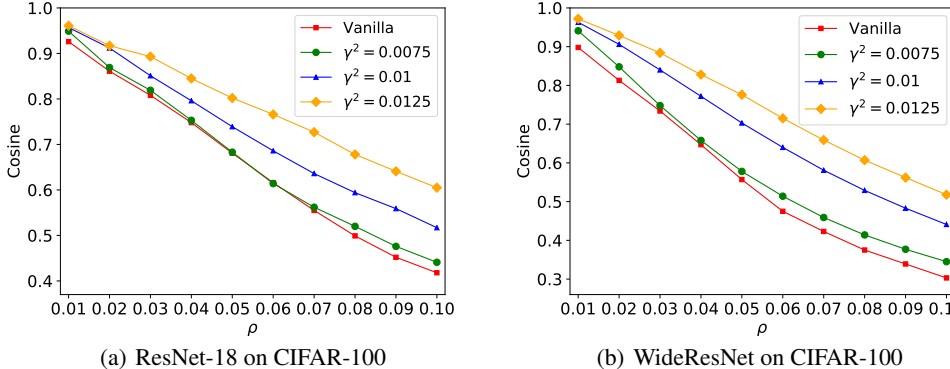

(a) ResNet-18 on CIFAR-100         (b) WideResNet on CIFAR-100

Figure 1: **Analysis about the Approximation of Inner Maximization.** We plot the cosine value between $g(w)$ and $g(w + \rho \frac{g(w)}{\|g(w)\|})$ (the red line) and between $g(\hat{w})$ and $g(\hat{w} + \rho \frac{g(\hat{w})}{\|g(\hat{w})\|})$ (the green, blue and orange line), where $g(w) = \nabla_w L(w)$ is the vanilla gradient, and $g(\hat{w}) = \nabla_{\hat{w}} L(\hat{w}) = \nabla_w L(w + \delta_0)$ represents the gradient on the weight with Gaussian initialization, where $\hat{w} = w + \delta_0$. Based on the analysis in [2], we use training-time BatchNorm to conduct the experiments. $\gamma$ is the standard variation of $\delta_0$.

**Random smoothing improves gradient stability.** Some studies illustrate that random smoothing can smooth the loss landscape [22, 46]. That motivates us to analyze the effects of random smoothing for the smoothness of gradient $g(w)$. In Figure 1, we sample a Gaussian noise $\delta_0$ from $\mathbb{N}(0, \gamma^2 I)$ and obtain the weight $\hat{w} = w + \delta_0$. Next, we calculate the cosine value between $g(\hat{w})$ and $g(\hat{w} + \rho \frac{g(\hat{w})}{\|g(\hat{w})\|})$ to compare the smoothness with the gradient without random smoothing ($g(w)$). From Figure 1, we can find that the cosine value with random smoothing (the green, blue and orange line) is higher than the vanilla cosine value on $g(w)$ (the red line), which means that random smoothing can reduce the decay rate of cosine value. In addition, the cosine value will be more stable with the $\gamma$ value increasing when using random smoothing (from $\gamma^2 = 0.0075$ to $\gamma^2 = 0.0125$) in Figure 1. The above observation illustrates that random smoothing can smooth the loss landscape and obtain a more stable gradient. We also provide the theoretical analysis for this observation.

**Theorem 1.** *Let $L(w)$ be the vanilla loss function and $L_S(w) = L(w + \delta_0)$ be the smoothed loss function. Assuming vanilla loss function $L(w)$ is $\alpha$-Lipschitz continuous and the gradient $\nabla_w L(w)$ is $\beta$-Lipschitz continuous. Given the random noise $\delta_0$ is from Gaussian distribution $\mathbb{N}(0, \gamma^2 I)$. We can obtain that the gradient $\nabla_w L_S(w) = \nabla_w L(w + \delta_0)$ is $\min\{\frac{\alpha}{\gamma}, \beta\}$-Lipschitz continuous, where $\gamma$ is the standard deviation of $\delta_0$.*

Theorem 1 is based on [5, 16], which implies that $L_S(w) = L(w + \delta_0)$ is more smooth than the original loss function $L(w)$ when $\frac{\alpha}{\gamma} \leq \beta$. In addition, with $\gamma$ value (standard deviation of Gaussian noise) increasing, the loss function $L_S(w)$ become smoother. That is also consistent with our observation in Figure 1, where the change of cosine value will be reduced with $\gamma^2$ increasing. Therefore, based on the above intuition about random smoothing, we can use the Gaussian noise $\delta_0$ to initialize the one-step gradient ascent process. That means we can firstly use Gaussian noise to escape from the local sharp region and smooth the loss landscape (initial perturbation), and then conduct the one-step gradient ascent step. More specifically, the random initialization for inner maximization can be defined as $\hat{w} = w + \delta_0$. After that, we use the smoothed weight $\hat{w}$ to achieve the one-step gradient ascent process and obtain the approximation of optimal adversarial weight:

$$w_{adv} = \hat{w} + \rho \frac{g(\hat{w})}{\|g(\hat{w})\|} = w + \delta_0 + \rho \frac{g(w + \delta_0)}{\|g(w + \delta_0)\|}. \tag{6}$$

Finally, we investigate whether the smoothed gradient can benefit the inner maximization and obtain a better approximated solution in practical applications. We try to plot the perturbed loss value ($L(w + \rho \frac{g(w)}{\|g(w)\|})$ and $L(w + \delta_0 + \rho \frac{g(w+\delta_0)}{\|g(w+\delta_0)\|})$) in Figure 2. From this figure, we can find that the loss value is very close for different $\gamma^2$ value when the $\rho$ value is small ($\rho \leq 0.2$). However, with

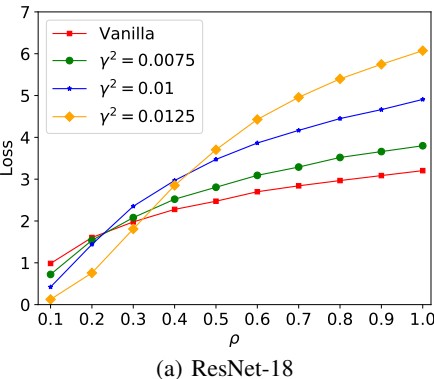
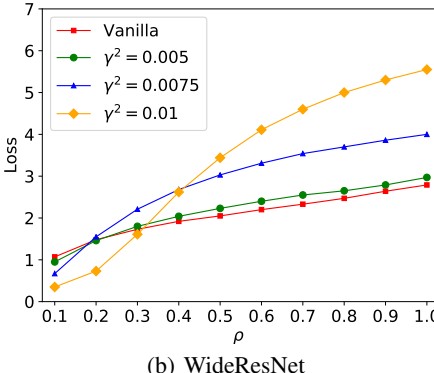

(a) ResNet-18               (b) WideResNet

Figure 2: **The effect of Gaussian noise for loss value.** The red line (vanilla) represents the loss function $L(\boldsymbol{w} + \rho \frac{\boldsymbol{g}(\boldsymbol{w})}{\|\boldsymbol{g}(\boldsymbol{w})\|})$ with $\rho$ value increasing. The other three lines show the change of loss function with random initialization $L(\hat{\boldsymbol{w}} + \rho \frac{\boldsymbol{g}(\hat{\boldsymbol{w}})}{\|\boldsymbol{g}(\hat{\boldsymbol{w}})\|})$ when increasing $\rho$ value, where $\hat{\boldsymbol{w}} = \boldsymbol{w} + \boldsymbol{\delta_0}$. Noted that $\boldsymbol{\delta_0}$ is from the Gaussian distribution with three different standard deviation ($\gamma^2 = 0.0075, 0.01, 0.0125$). We select the checkpoint of 200-th epoch on CIFAR-100 as the weight $\boldsymbol{w}$ to plot the figure.

the $\rho$ value gradually increasing, the loss value grow slowly for vanilla weight $\boldsymbol{w}$. That means $\boldsymbol{g}(\boldsymbol{w})$ cannot efficiently increase the loss value and is not applicable for solving the inner maximization with the region expanding. That limited the region size that vanilla weight can select. In addition, we observe that the weight with Gaussian noise ($\boldsymbol{w} + \boldsymbol{\delta_0}$) shows a more stable pattern for the increase of loss value to solve the inner maximization problem. Applying one-step gradient ascent based on the gradient of smoothed weight $\boldsymbol{w} + \boldsymbol{\delta_0}$ can better maximize the loss value in the neighbouring region. The above analysis motivates us to propose a random smoothing based SAM algorithm to obtain a better solution for inner maximization. We will introduce our proposed Random Sharpness-Aware Minimization algorithm (R-SAM) in the next section.

### 3.2 Random Sharpness-Aware Minimization (R-SAM)

Based on the observation and intuition in section 3.1, we propose a novel random initialization based Sharpness-Aware Minimization algorithm (R-SAM) to benefit the solution of inner maximization in SAM. For R-SAM, we use Gaussian noise to initialize the perturbation for weight $\boldsymbol{w}$ and obtain the smoothed weight $\hat{\boldsymbol{w}} = \boldsymbol{w} + \boldsymbol{\delta_0}$. After that, we use the smoothed weight $\hat{\boldsymbol{w}}$ to calculate the gradient $\boldsymbol{g}(\hat{\boldsymbol{w}})$ for gradient ascent process: $\boldsymbol{g}(\hat{\boldsymbol{w}}) = \nabla_{\hat{\boldsymbol{w}}} L(\hat{\boldsymbol{w}}) = \nabla_{\boldsymbol{w}} L(\boldsymbol{w} + \boldsymbol{\delta_0})$. In this way, we can obtain a more stable gradient to solve the inner maximization problem. More specifically, we will use the gradient $\boldsymbol{g}(\hat{\boldsymbol{w}})$ to finish the gradient ascent step and obtain adversarial weight $\boldsymbol{w_{adv}}$. A simple way to obtain $\boldsymbol{w_{adv}}$ is that we can directly add the gradient-based perturbation $\boldsymbol{g}(\hat{\boldsymbol{w}}) = \nabla_{\hat{\boldsymbol{w}}} L(\hat{\boldsymbol{w}})$ to the smoothed weight $\hat{\boldsymbol{w}} = \boldsymbol{w} + \boldsymbol{\delta_0}$, which is defined as $\boldsymbol{w_{adv}} = \boldsymbol{w} + \boldsymbol{\delta_0} + \rho \frac{\boldsymbol{g}(\hat{\boldsymbol{w}})}{\|\boldsymbol{g}(\hat{\boldsymbol{w}})\|}$. However, that will cause the norm of total perturbation $\boldsymbol{\delta_0} + \rho \frac{\boldsymbol{g}(\hat{\boldsymbol{w}})}{\|\boldsymbol{g}(\hat{\boldsymbol{w}})\|}$ larger than vanilla $\rho$. To constrain the norm of total perturbation, we try to project the perturbation into the ball with radius $\rho$. The final perturbation can be defined as:

$$\boldsymbol{w_{adv}} = \boldsymbol{w} + \rho \frac{\boldsymbol{\delta_0} + \lambda \boldsymbol{g}(\hat{\boldsymbol{w}})}{\|\boldsymbol{\delta_0} + \lambda \boldsymbol{g}(\hat{\boldsymbol{w}})\|} \tag{7}$$

where $\boldsymbol{\delta_0}$ represents the Gaussian noise for random initialization and $\beta$ is used to control the ratio between $\boldsymbol{\delta_0}$ and $\boldsymbol{g}(\hat{\boldsymbol{w}})$.

Noted that the selection of Gaussian noise plays an important role in R-SAM. However, unlike the input image data that each pixel is from the same distribution, the distribution of weights in each layer or kernel shows different distributions. That makes the noise design a challenging problem. Related

**Algorithm 1** Random-SAM (R-SAM)

---

**Input:** $x \in \mathbb{R}^d$, $\boldsymbol{w}$ represents vanilla weight, $\hat{\boldsymbol{w}}$ is the smoothed weight, learning rate $\eta_t$, $k$ is the number of kernels in model.

**for** $t \leftarrow 1$ **to** $T$ **do**

    Sample Minibatch $\mathcal{B} = \{(x_i, y_i), \cdots, (x_{|\mathcal{B}|}, y_{|\mathcal{B}|})\}$ from $X$.

    $\boldsymbol{\delta_0} = \mathbb{N}(0, \gamma^2 \cdot diag(||\boldsymbol{w_j}||_{j=1}^k))$

    Obtain a stable gradient $\boldsymbol{g}(\hat{\boldsymbol{w}}) = \nabla_{\hat{\boldsymbol{w}}} L(\hat{\boldsymbol{w}}) = \nabla_{\boldsymbol{w}} \mathcal{L}(\boldsymbol{w} + \boldsymbol{\delta_0})$ on minibatch $\mathcal{B}$.

    Compute gradient ascent $\boldsymbol{w_{adv}} = \boldsymbol{w} + \rho \frac{\boldsymbol{\delta_0} + \lambda \boldsymbol{g}(\hat{\boldsymbol{w}})}{||\boldsymbol{\delta_0} + \lambda \boldsymbol{g}(\hat{\boldsymbol{w}})||}$

    Compute SAM gradient: $\boldsymbol{g_{sam}} = \nabla_{\boldsymbol{w}} L(\boldsymbol{w})|_{\boldsymbol{w_{adv}}}$

    Update weights: $\boldsymbol{w_{t+1}} = \boldsymbol{w_t} - \eta_t \boldsymbol{g_{sam}}$

**end for**

---

work has illustrated that the weight in each kernel is from an independent Gaussian distribution [25]. Inspired by this work, in this paper, we use kernel-wise Gaussian noise $\boldsymbol{\delta_0} \sim \mathbb{N}(0, \gamma^2 \cdot diag(||\boldsymbol{w_j}||_j^k))$ as the initialization of perturbation in R-SAM, where $||\boldsymbol{w_j}||_j^k$ represents the $\ell_2$-norm of $k$-th kernel in the neural network.

The pseudo-code of R-SAM can be found in Algorithm 1. Firstly, we need to sample a minibatch data $\mathcal{B} = \{(x_i, y_i), \cdots, (x_{|\mathcal{B}|}, y_{|\mathcal{B}|})\}$ from dataset $X$. After that, we will calculate the norm of each kernel $||\boldsymbol{w_j}||_{j=1}^k$ and sample kernel-wise Gaussian noise from the distribution $\mathbb{N}(0, \gamma^2 \cdot diag(||\boldsymbol{w_j}||_j^k))$ as the initialization of perturbation. Then we can obtain $\boldsymbol{g}(\hat{\boldsymbol{w}}) = \nabla_{\boldsymbol{w}} \mathcal{L}(\boldsymbol{w} + \boldsymbol{\delta_0})$ as the stable gradient to better approximate the optimal adversarial weight $\boldsymbol{w_{adv}^*}$. More specifically, we can use one-step gradient ascent to calculate $\boldsymbol{w_{adv}} = \boldsymbol{w} + \rho \frac{\boldsymbol{\delta} + \lambda \boldsymbol{g}(\hat{\boldsymbol{w}})}{||\boldsymbol{\delta} + \lambda \boldsymbol{g}(\hat{\boldsymbol{w}})||}$. In this way, we can calculate the gradient on $\boldsymbol{w_{adv}}$: $\boldsymbol{g_{sam}} = \nabla_{\boldsymbol{w}} L(\boldsymbol{w})|_{\boldsymbol{w_{adv}}}$. Finally, following the process in vanilla SAM, we can use the gradient $\boldsymbol{g_{sam}}$ to update the vanilla weight $\boldsymbol{w}$. As shown in Algorithm 1, R-SAM is very easy to reproduce.

## 4 Experiments

### 4.1 Experimental Settings

In this section, we try to evaluate the performance of our proposed R-SAM optimizer on ResNet [23], WideResNet [48] with CIFAR [33] datasets and Vision Transformer (ViT) [14] with ImageNet [11] dataset. First, compared with vanilla SAM, the experimental results illustrate that our proposed R-SAM can improve the accuracy in CIFAR and ImageNet. In addition, the ablation study and sensitivity analysis show that our proposed R-SAM is robust for hyper-parameters.

### 4.1.1 Datasets

To validate the performance of proposed method R-SAM, we firstly try to conduct the experiments on widely used CIFAR-10, CIFAR-100 [33] and ImageNet-1k [11] datasets. To further validate the performance and robustness of R-SAM, we also report the experiment results on ImageNet-Real [4], ImageNet-V2 [41], ImageNet-R [24] and ImageNet-C [19].

### 4.1.2 Models

We firstly evaluate the performance of our proposed R-SAM in CNN-based models, such as ResNet-18, ResNet-50 [23] and WideResNet (WRN-28-20) [48]. Currently, Transformer-based models receive a great deal of attention in the field of computer vision [37, 43]. After that, we also try to evaluate our proposed algorithm R-SAM in Vision Transformer (ViT) [14].

### 4.1.3 Baselines

To illustrate that the random initialization in R-SAM can help to escape the non-smooth vicinity and benefit to the approximation in inner maximization, the main baseline of this paper is vanilla SAM [18]. In addition, to show that SGD+M is more suitable for CIFAR training, we also try to compare

the accuracy of SGD+M with SGD, RMSProp and AdamW [38]. To further show the generality of R-SAM, we also try to combine it with GSAM [53] and propose R-GSAM algorithm. The experiment results illustrate that R-GSAM can obtain a better performance compared with vanilla GSAM in ViT training. That also illustrates the effectiveness of random initialization in R-SAM and R-GSAM.

### 4.1.4 Implementation Details

The experiments in this paper are implemented with JAX [6] on Google TPU-V3 chips. The hyper-parameters and experimental setting is based on vanilla SAM [8, 18]. For ResNet training, we use SGD with Momentum as the base optimizer of SAM and R-SAM. For ViT training, the base optimizer of SAM and R-SAM is AdamW [38]. As for data augmentation, we select mixup [49] and RandAugment [10] to conduct the experiments. Noted that we do not use stochastic depth for ViT training. The implementation details can be found in Appendix 7.4.

## 4.2 Training from Scratch on CIFAR-10

To evaluate the performance of our proposed R-SAM, we conduct experiments of ResNet-18, ResNet-50 and WideResNet (WRN-28-20) training on CIFAR-10 dataset. Following the experimental setting in vanilla SAM [18], we use basic augmentation to preprocess the input image data and the base optimizer for SAM and R-SAM is SGD with Momentum (SGD+M).

The experimental results are shown in Table 1. We can find that SAM-based optimizers (SAM and R-SAM) can achieve a better performance than traditional first-order optimizers (SGD and AdamW). For example, the accuracy of traditional first-order optimizers are around 96%, and the best accuracy is 96.5% from SGD with Momentum (SGD+M). However, SAM can improve the accuracy from 96.5% to 97.3% for WRN-28-20. That illustrates the power of SAM to improve the performance of neural networks. In addition, R-SAM can further improve the performance compared with vanilla SAM. More specifically, R-SAM can improve the accuracy of SAM from 97.3% to 97.5% on WRN-28-20 for CIFAR-10. Noted that the accuracy of SAM for CIFAR-10 is pretty high and the improvement of R-SAM is also convincing.

Table 1: Accuracy of ResNet and WideResNet on CIFAR-10 for 200 epoch. The base optimizer for SAM and R-SAM is SGD with Momentum (SGD+M). We select batch size as 128.

| Model | SGD | SGD+M | RMSProp | AdamW | LPF-SGD | SAM | R-SAM |
|---|---|---|---|---|---|---|---|
| **ResNet-18** | 95.4±0.1 | 95.6±0.2 | 95.4±0.2 | 95.1±0.1 | 95.9±0.1 | 96.4±0.2 | **96.5**±0.1 |
| **ResNet-50** | 95.8±0.1 | 95.7±0.1 | 95.7±0.1 | 96.0±0.1 | 96.3±0.2 | 96.7±0.1 | **96.9**±0.1 |
| **WRN-28-10** | 96.4±0.1 | 96.5±0.1 | 96.4±0.2 | 96.0±0.1 | 96.8±0.1 | 97.3±0.1 | **97.5**±0.1 |

## 4.3 Training from Scratch on CIFAR-100

To further validate the performance of our proposed R-SAM, we also try to evaluate its accuracy on CIFAR-100 dataset. We also follow the setting in SAM [18] and use basic augmentation to preprocess the image data. We select SGD+M as the base optimizer of SAM and R-SAM.

Table 2: Accuracy of ResNet and WideResNet on CIFAR-100 for 200 epoch. The base optimizer for SAM and R-SAM is SGD with Momentum (SGD+M). We select batch size as 128.

| Model | SGD | SGD+M | RMSProp | AdamW | LPF-SGD | SAM | R-SAM |
|---|---|---|---|---|---|---|---|
| **ResNet-18** | 78.0±0.1 | 78.9±0.2 | 79.4±0.1 | 77.7±0.1 | 80.2±0.1 | 80.9±0.2 | **81.4**±0.2 |
| **ResNet-50** | 80.9±0.2 | 81.4±0.2 | 81.4±0.1 | 80.8±0.1 | 82.1±0.2 | 83.3±0.1 | **84.0**±0.1 |
| **WRN-28-10** | 81.1±0.1 | 81.7±0.1 | 81.7±0.1 | 80.1±0.2 | 82.6±0.1 | 84.6±0.1 | **85.2**±0.1 |

The experimental results are shown in Table 2, we notice that the results show a similar pattern with CIFAR-10 and can achieve a higher improvement. In particular, SAM-based optimizers can obtain better performance compared with first-order optimizers. For example, the best accuracy of first-order

optimizers is 81.7% for WRN-28-20 model. SAM can improve the accuracy from 81.7% to 84.6% for WRN-28-10 and from 81.4% to 83.3% for ResNet-50. In addition, R-SAM can obtain a higher accuracy compared (from 84.6% to 85.2% for WRN-28-20 and 83.3% to 84.0% for ResNet-50). The above experimental results also illustrate that our proposed R-SAM can further help SAM to obtain better performance.

## 4.4 ViT Training from Scratch on ImageNet

Currently, Vision Transformer has been widely used in a variety of areas [37, 43]. Chen et al. illustrate that SAM can significantly improve the performance of ViT training. Therefore, evaluating the performance of our proposed R-SAM on ViT is pretty important for verifying our contribution. In this section, we select ViT-B-16 and ViT-S-16 to evaluate the performance of R-SAM. The experimental setting follows the introduction in [8, 18]. Noted that the base optimizer of SAM and R-SAM are both AdamW. We select batch size as 4096 for ViT training.

**ViT Training.** The experimental results of ViT training on ImageNet are shown in Table 3. From this table, we can observe that SAM can also significantly improve the accuracy of ViT training. For example, the accuracy of ViT-B-16 is improved from 74.7% (AdamW) to 79.8% (SAM). In addition, when using SAM, the accuracy of ViT-S-16 can also be improved from 74.9% to 77.9%. However, our proposed R-SAM can further improve the accuracy of ViT training compared with vanilla SAM and achieve better performance. More specifically, R-SAM can improve the accuracy of ViT-B-16 from 79.8% (vanilla SAM) to 80.7%. Besides, ViT-S-16 can also obtain benefits from R-SAM and the accuracy is increased from 77.9% (vanilla SAM) to 78.7%. The above experimental results further validate the effectiveness of our proposed R-SAM, especially when compared with vanilla SAM.

Table 3: Accuracy of ViT on ImageNet-1k for 300 epoch. The base optimizer for SAM and R-SAM is AdamW. Batch Size is 4096. We use Inception-style preprocessing method for input image.

| Model | Resolution | Mixup | RandAug | AdamW | SAM | R-SAM |
|-------|------------|-------|---------|-------|-----|-------|
| **ViT-B-16** | 224 | | | 74.7 | 79.8 | **80.7** |
| **ViT-S-16** | 224 | | | 74.9 | 77.9 | **78.7** |
| **ViT-B-16** | 224 | ✓ | | 75.7 | 80.4 | **81.1** |
| **ViT-B-16** | 224 | ✓ | ✓ | 79.6 | 80.8 | **81.6** |

**Data Augmentation.** To further evaluate the generality of our proposed R-SAM, we also try to combine R-SAM with strong augmentation methods and the results are shown in Table 3. In this experiment, we select mixup [49] and RandAugment (RandAug) [10] as the augmentation methods to conduct the experiments. Firstly, we can find that augmentation methods can further improve the accuracy of AdamW and vanilla SAM. For instance, we can observe that mixup can improve the accuracy of vanilla ViT-B-16 with AdamW from 74.7% to 75.7%. In addition, when SAM is selected, the accuracy will further increase from 79.8% to 80.4% with mixup for ViT-B-16.

In addition, the experimental results illustrate that our proposed R-SAM can further improve the accuracy of vanilla SAM even with the strong augmentations. For example, R-SAM can improve the accuracy of SAM from 80.4% to 81.1% even using strong augmentations. These experiments further verify that our proposed R-SAM can be combined with strong augmentations and illustrate a great generality. Finally, we find that mixup and RandAug can work together to benefit the performance of R-SAM. As shown in Table 3, when RandAug is selected, the accuracy is increased to 81.6% compared to simply using only the mixup.

**Further Evaluations and Robustness.** Firstly, to further validate the performance of R-SAM, we report the experiment results on ImageNet-Real and ImageNet-v2. As shown in Table 4, we can observe that R-SAM can still improve the accuracy of ViT compared with vanilla SAM. For example, R-SAM can improve the accuracy of ViT-B-16 from 85.2% to 86.1% on ImageNet-Real. In addition, to evaluate the robustness of trained model with R-SAM, we also conduct the experiments in ImageNet-R and ImageNet-C datasets. The experimental results are also shown in Table 4 , which illustrate that SAM can improve the robustness of trained model compared with AdamW. In addition, R-SAM can maintain the advantage of SAM and further strengthen its robustness. For example, SAM

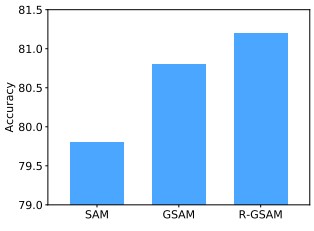 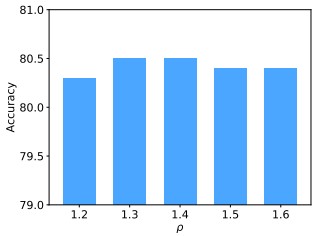 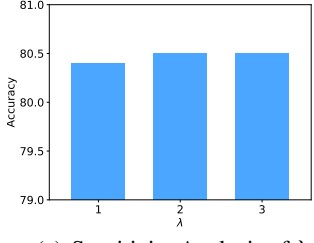

(a) Comparison with GSAM     (b) Sensitivity Analysis of $\rho$     (c) Sensitivity Analysis of $\lambda$

Figure 3: (a) **Comparison with GSAM.** Vanilla R-SAM can achieve similar accuracy for ViT-B-16 training. R-GSAM can obtain a higher accuracy. (b) **Sensitivity Analysis of R-SAM about $\rho$.** We can find that R-SAM shows a stable pattern with different $\rho$ value. (c) **Sensitivity Analysis of R-SAM about $\lambda$.** R-SAM can obtain a stable improvement under different $\lambda$ value.

Table 4: Evaluation of R-SAM on ImageNet datasets for 300 epoch. The base optimizer for SAM and R-SAM is AdamW. We select batch size as 4096. We also use Inception-style preprocessing method for input image.

| Model | Training | ImageNet | ImageNet-Real | ImageNet-v2 | ImageNet-R | ImageNet-C |
|-------|----------|----------|---------------|-------------|------------|------------|
| **ViT-B-16** | AdamW | 74.6 | 79.8 | 61.3 | 20.1 | 46.6 |
| | SAM | 79.8 | 85.2 | 67.5 | 26.4 | 56.5 |
| | R-SAM | **80.7** | **86.1** | **68.6** | **26.9** | **56.8** |
| **ViT-S-16** | AdamW | 74.4 | 80.4 | 61.7 | 20.0 | 46.5 |
| | SAM | 77.9 | 84.1 | 65.6 | 24.7 | 53.0 |
| | R-SAM | **78.7** | **84.9** | **66.7** | **25.1** | **53.2** |

can improve the accuracy of ViT-B-16 from 20.1% to 26.4% on ImageNet-R. R-SAM can further improve the accuracy to 26.9%.

## 4.5 Compared with Algorithms in the Literature

In section 2.1, we introduced some related work about SAM. In this section, we try to evaluate the performance of R-SAM compared with the algorithms in the literature. We find that GSAM can achieve the best accuracy for ViT training. Therefore, we mainly focus on the comparison with GSAM. The main reults are from ViT-B-16 training on ImageNet-1k dataset and are shown in Figure 3(a). We can find that R-SAM can obtain the similar accuracy compared with vanilla GSAM. Howvever, when combining R-SAM and GSAM and named it as R-GSAM, we can obtain better performance. For instance, the accuracy of R-SAM and GSAM are 80.7% and 80.8% respectively. When using R-SAM for GSAM and obtain the results of R-GSAM, we can observe that the accuracy increase from 80.8% to 81.2%. The main purpose of this paper is to illustrate that random smoothing can benefit the inner maximization and improve the performance of SAM.

## 4.6 Sensitivity Analysis

Reproducibility is critical for deep learning applications. We notice that deep learning is usually sensitive to hyper-parameters. Therefore, in this section, we try to study the effects of hyperparameters for the performance of ViT training on ImageNet-1k dataset. More specifically, we mainly focus on the effects of $\rho$ and $\lambda$. The reason is that $\rho$ determines the radius of perturbation and $\lambda$ controls the ratio between Gaussian noise $\boldsymbol{\delta_0}$ and gradient-based perturbation $\frac{\boldsymbol{g}(\hat{\boldsymbol{w}})}{\|\boldsymbol{g}(\hat{\boldsymbol{w}})\|}$. The analysis results are shown in Figure 3. We can find that $\rho$ and $\lambda$ shows a stable pattern for tuning. That further improves the reproducibility of R-SAM and make it more applicable in real scenarios.

## 5 Conclusion

In this paper, we observe that simply applying one-step gradient ascent in the inner maximization of SAM may suffer from poor approximation error due to unstable gradient. To resolve this problem, we propose a novel algorithm R-SAM, which provably improves the gradient stability by initializing the inner maximization of SAM with a Gaussian noise. The experimental results on ResNet and ViT illustrate that R-SAM can improve the generalization performance of the model and yield better performance compared with SAM on CIFAR-10, CIFAR-100 and ImageNet datasets.

## 6 Acknowledgements and Disclosure of Funding

We thank Google TFRC for supporting us to get access to the Cloud TPUs. We thank CSCS (Swiss National Supercomputing Centre) for supporting us to get access to the Piz Daint supercomputer. We thank TACC (Texas Advanced Computing Center) for supporting us to get access to the Longhorn supercomputer and the Frontera supercomputer. We thank LuxProvide (Luxembourg national super-computer HPC organization) for supporting us to get access to the MeluXina supercomputer. CJH and XC are partially supported by NSF IIS-2008173, NSF IIS-2048280 and research awards from Google, Sony, Samsung and Okawa Foundation. YY and YL are also partially supported by grants from Alibaba, Bytedance, Huawei, and Singapore Maritime Institute.

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
