# 7 Appendix

## 7.1 Proof of Theorem 1

The Proof is based on [5, 16]. Before we prove the theorem, we need the following lemma.

**Lemma 1.** *(Bisla et al.) Let $L(w)$ be $\alpha$-Lipschitz continuous. Let $\delta$ be distributed according to the distribution $p$. Then,*

$$
\begin{aligned}
\|\nabla L_S(x) - \nabla L_S(y)\| &= \mathbb{E}_{\delta \sim p}[\nabla L_S(x + \delta) - \nabla L_S(y + \delta)] \\
&\leq \alpha \int |p(\delta - x) - p(\delta - y)| d\delta.
\end{aligned}
\tag{8}
$$

Then we provide the proof of Theorem 1. We firstly illustrate that $\nabla L_S(w)$ is $\frac{\alpha}{\gamma}$-Lipschitz continuous, and then demonstrate that $\nabla L_S(w)$ is $\beta$-Lipschitz continuous. Finally, we can obtain that $\nabla L_S(w)$ is $\min\{\frac{\alpha}{\gamma}, \beta\}$-Lipschitz continuous.

We can define $\int |p(\delta - x) - p(\delta - y)| d\delta$ as:

$$
\begin{aligned}
&\int |p(\delta - x) - p(\delta - y)| d\delta \\
&= \int_{\delta:\|\delta-x\|\geq\|\delta-y\|} [p(\delta - x) - p(\delta - y)] d\delta + \int_{\delta:\|\delta-x\|\leq\|\delta-y\|} [p(\delta - y) - p(\delta - x)] d\delta \\
&= 2 \int_{\delta:\|\delta-x\|\leq\|\delta-y\|} [p(\delta - x) - p(\delta - y)] d\delta.
\end{aligned}
\tag{9}
$$

We try to divide $\int |p(\delta - x) - p(\delta - y)| d\delta$ into two different parts in the first equality. The second equality holds because they have the equal value when $\|p(\delta - x)\| \geq \|p(\delta - y)\|$ and $\|p(\delta - x) \leq p(\delta - y)\|$.

Let $\hat{\delta} = \delta - x$ for $p(\delta - x)$ term and $\hat{\delta} = \delta - y$ for $p(\delta - y)$ term, we have:

$$
\begin{aligned}
&\int |p(\delta - x) - p(\delta - y)| d\delta \\
&= 2 \int_{\delta:\|\delta-x\|\leq\|\delta-y\|} p(\delta - x) d\delta - 2 \int_{\delta:\|\delta-x\|\leq\|\delta-y\|} p(\delta - y) d\delta \\
&= 2 \int_{\delta:\|\hat{\delta}\|\leq\|\hat{\delta}+(x-y)\|} p(\hat{\delta}) d\delta - 2 \int_{\delta:\|\hat{\delta}\|\geq\|\hat{\delta}-(x-y)\|} p(\hat{\delta}) d\delta.
\end{aligned}
\tag{10}
$$

We can rewrite Equation 10 as:

$$
\begin{aligned}
&\int |p(\delta - x) - p(\delta - y)| d\delta \\
&= 2\mathbb{P}_{\delta \sim p}(\|\delta\| \leq \|\delta + (x - y)\|) - 2\mathbb{P}_{\delta \sim p}(\|\delta\| \geq \|\delta - (x - y)\|).
\end{aligned}
\tag{11}
$$

For the first part of the Equation 11, we can find:

$$
\begin{aligned}
&\mathbb{P}_{\delta \sim p}(\|\delta\| \leq \|\delta + (x - y)\|) \\
&= \mathbb{P}_{\delta \sim p}(\|\delta\|^2 \leq \|\delta + (x - y)\|^2) \\
&= \mathbb{P}_{\delta \sim p}(2\langle \delta, x - y \rangle \geq -\|x - y\|^2) \\
&= \mathbb{P}_{\delta \sim p}(2\langle \delta, \frac{x - y}{\|x - y\|} \rangle \geq -\|x - y\|).
\end{aligned}
\tag{12}
$$

In addition, $\frac{x-y}{\|x-y\|}$ has norm 1 and $\delta \sim \mathbb{N}(0, \gamma^2 I)$ implies $\langle \delta, \frac{x-y}{\|x-y\|} \rangle \sim \mathbb{N}(0, \gamma^2 I)$. Therefore,

$$
\begin{aligned}
&\mathbb{P}_{\delta \sim p}(\|\delta\| \leq \|\delta + (x-y)\|) \\
=&\mathbb{P}_{\delta \sim p}(\langle \delta, \frac{x-y}{\|x-y\|} \rangle \geq -\frac{\|x-y\|}{2}) \\
=&\int_{-\frac{\|x-y\|}{2}}^{+\infty} \frac{1}{\sqrt{2\pi\gamma^2}} \exp(-\frac{\hat{\delta}^2}{2\gamma^2})d\hat{\delta}.
\end{aligned}
\tag{13}
$$

For the second part in Equation 11, we can obtain that:

$$
\begin{aligned}
&\mathbb{P}_{\delta \sim p}(\|\delta\| \geq \|\delta - (x-y)\|) \\
=&\mathbb{P}_{\delta \sim p}(\|\delta\|^2 \geq \|\delta - (x-y)\|^2) \\
=&\mathbb{P}_{\delta \sim p}(2\langle \delta, x-y \rangle \geq \|x-y\|^2) \\
=&\mathbb{P}_{\delta \sim p}(2\langle \delta, \frac{x-y}{\|x-y\|} \rangle \geq \|x-y\|).
\end{aligned}
\tag{14}
$$

We can also get the similar distribution $\langle \delta, \frac{x-y}{\|x-y\|} \rangle \sim \mathbb{N}(0, \gamma^2 I)$. Therefore, we can find:

$$
\begin{aligned}
&\mathbb{P}_{\delta \sim p}(\|\delta\| \geq \|\delta - (x-y)\|) \\
=&\mathbb{P}_{\delta \sim p}(\langle \delta, \frac{x-y}{\|x-y\|} \rangle \geq \frac{\|x-y\|}{2}) \\
=&\int_{\frac{\|x-y\|}{2}}^{+\infty} \frac{1}{\sqrt{2\pi\gamma^2}} \exp(-\frac{\hat{\delta}^2}{2\gamma^2})d\hat{\delta}.
\end{aligned}
\tag{15}
$$

Then, we can combine Equation 13 and 15 to Equation 11:

$$
\begin{aligned}
&\int |p(\delta - x) - p(\delta - y)|d\delta \\
=& 2\int_{-\frac{\|x-y\|}{2}}^{+\infty} \frac{1}{\sqrt{2\pi\gamma^2}} \exp(-\frac{\hat{\delta}^2}{2\gamma^2})d\hat{\delta} - 2\int_{\frac{\|x-y\|}{2}}^{+\infty} \frac{1}{\sqrt{2\pi\gamma^2}} \exp(-\frac{\hat{\delta}^2}{2\gamma^2})d\hat{\delta} \\
=& 2\int_{-\frac{\|x-y\|}{2}}^{\frac{\|x-y\|}{2}} \frac{1}{\sqrt{2\pi\gamma^2}} \exp(-\frac{\hat{\delta}^2}{2\gamma^2})d\hat{\delta} \\
\leq& \frac{\sqrt{2}\|x-y\|}{\gamma\sqrt{\pi}}
\end{aligned}
\tag{16}
$$

Therefore, we can combine Equation 8 and Equation 16:

$$
\begin{aligned}
\|\nabla L_S(x) - \nabla L_S(y)\| \leq& \alpha \int |p(\delta - x) - p(\delta - y)|d\delta \\
\leq& \alpha \frac{\sqrt{2}\|x-y\|}{\gamma\sqrt{\pi}} \\
\leq& \frac{\alpha}{\gamma}\|x-y\|
\end{aligned}
\tag{17}
$$

Finally, we finish the proof of $\frac{\alpha}{\gamma}$-Lipschitz continuous for $\nabla L_S(w)$. Then, we try to show the proof of $\beta$-Lipschitz continuous for $\nabla L_S(w)$:

$$\|\nabla L_S(x) - \nabla L_S(y)\| = \|\nabla \mathbb{E}_{\delta \sim p}[L(x+\delta)] - \nabla \mathbb{E}_{\delta \sim p}[L(x+\delta)]\|$$
$$= \|\mathbb{E}_{\delta \sim p}[\nabla L(x+\delta)] - \nabla L(y+\delta)\|$$
$$= \|\int [\nabla L(x+\delta)] - \nabla L(y+\delta)]p(\delta)d\delta\|$$
$$\leq \int \|\nabla L(x+\delta) - \nabla L(y+\delta)\|p(\delta)d\delta \tag{18}$$
$$\leq \int \beta\|(x+\delta)\| \int p(\delta)d\delta$$
$$= \beta\|x-y\| \int p(\delta)d\delta$$
$$= \beta\|x-y\|$$

Therefore, we can obtain $\nabla L_S(w)$ is $\beta$-Lipschitz continuous.

Finally, we finish the proof that $\nabla L_S(w)$ is $\min\{\frac{\alpha}{\gamma}, \beta\}$-Lipschitz continuous.

## 7.2 Compared with LPF-SGD

As shown in Table 1 and Table 2, we try to compare R-SAM with LPF-SGD on CIFAR-10 and CIFAR-100. In this section, we also try to compare R-SAM with LPF-SGD on ImageNet.

Table 5: Accuracy of ViT on ImageNet-1k for 300 epoch. The base optimizer for SAM and R-SAM is AdamW. Batch Size is 4096. We use Inception-style preprocessing method for input image.

| Model | Resolution | AdamW | LPF-SGD | SAM | R-SAM |
|---|---|---|---|---|---|
| **ViT-B-16** | 224 | 74.7 | 75.9 | 79.8 | **80.7** |
| **ViT-S-16** | 224 | 74.9 | 75.8 | 77.9 | **78.7** |

## 7.3 The sensitivity Analysis of $\gamma$

In this section, we try to analyze the sensitivity of $\gamma$ for R-SAM. The result is shown in Table 6.

Table 6: The sensitivity Analysis about $\gamma$

| Model | 2e-3 | 1.5e-3 | 1e-3 | 5e-4 | 1e-4 | 5e-5 |
|---|---|---|---|---|---|---|
| **WRN-28-10** | 84.7 | 85.1 | 85.3 | 85.4 | 85.1 | 84.5 |
| **ViT-B-16** | 79.7 | 80.2 | 80.7 | 80.7 | 80.4 | 79.7 |
| **ViT-S-16** | 77.3 | 77.9 | 78.5 | 78.2 | 77.9 | 77.8 |

## 7.4 Hyperparameters

Table 7: Architectures of Vision Transformer

| Model | Params | Patch Resolution | Sequence Length | Hidden Size | Heads | Layers |
|---|---|---|---|---|---|---|
| ViT-B-16 | 87M | $16 \times 16$ | 196 | 768 | 12 | 12 |
| ViT-S-16 | 22M | $16 \times 16$ | 196 | 384 | 6 | 12 |

Table 8: Parameter Settings of ViT from Scratch on ImageNet

| Model | Input Resolution | Batch Size | Epoch | Warmup Steps | Peak LR | LR Decay | Optimizer | $\rho$ | $\lambda$ | Weight Decay | Gradient Clipping |
|---|---|---|---|---|---|---|---|---|---|---|---|
| ViT-B-16 | 224 | 4096 | 300 | 10000 | 3e-3 | cosine | AdamW | / | / | 0.3 | 1.0 |
| ViT-S-16 | 224 | 4096 | 300 | 10000 | 3e-3 | cosine | AdamW | / | / | 0.3 | 1.0 |
| ViT-B-16 + SAM | 224 | 4096 | 300 | 10000 | 3e-3 | cosine | AdamW | 0.18 | / | 0.3 | 1.0 |
| ViT-S-16 + SAM | 224 | 4096 | 300 | 10000 | 3e-3 | cosine | AdamW | 0.1 | / | 0.2 | 1.0 |
| ViT-B-16 + R-SAM | 224 | 4096 | 300 | 10000 | 3e-3 | linear | AdamW | 1.4 | 2 | 0.3 | 1.0 |
| ViT-S-16 + R-SAM | 224 | 4096 | 300 | 10000 | 3e-3 | linear | AdamW | 0.4 | 1 | 0.2 | 1.0 |