# OpenReview forum: "Random Sharpness-Aware Minimization"
_NeurIPS.cc/2022/Conference — NeurIPS 2022 Accept_

### Official Review · Reviewer_Dv1G · 2022-07-05

**Rating:** 7
**Confidence:** 3
**Soundness:** 3 good
**Presentation:** 3 good
**Contribution:** 3 good

**Summary:**

The authors argue that the approximation of SAM's perturbation step, which requires solving an inner maximization objective, is prone to unstable gradients. To improve this approximation step without adding significant additional computational steps, the authors propose adding Gaussian noise to the perturbation initialization.

**Questions:**

Let me rephrase my previously mentioned weakness as a question here.

It is not entirely clear why performing the inner maximization gradient ascent step on a smoothed loss function necessarily leads to a more "accurate" solution or exactly what that means. Following the same argument, we would expect that we want to apply SGD (solving a minimization objective) on a smoothed loss function too. Yet, this is not standard practice. Works like [1] recently proposed to explore such smoothening; however, with the motivation of finding flat minima (akin to SAM), not "gradient stability" or more "accurate" solutions. It would be great if the authors could clarify why "gradient stability" is important for inner maximization.

I'm happy to raise my score if this gets sufficiently addressed.

[1] Low-Pass Filtering SGD for Recovering Flat Optima in the Deep Learning Optimization Landscape, Bisla et al, AISTATS 2022.

**Limitations:**

As far as I can see, the authors do not discuss any limitations of their method.


**Strengths And Weaknesses:**

# Strengths

* Significance: Efficient stochastic optimization is fundamental for deep learning, and sharpness-aware methods often (but not always [3]) lead to significantly better generalization performance than previous optimizers. Hence, improving methods like SAM can be very relevant to many optimizer users.
* Clarity: the paper is well-written and easy to follow.
* Simplicity: The proposed method is simple, easy to implement, and effective for the shown experiments shown.

# Weaknesses


While I'm open to being convinced otherwise and raising my score, I'm not fully convinced yet that the main premise of the paper -- randomly smoothening (RS) the loss function **significantly** **improves** over a non-smoothed loss function for approximating the gradient ascent step -- is true. RS seems very heuristical to me; hence, I'd appreciate either more clarification on why it always improves (if so) or more discussion on when it may not hold and what its trade-offs are.

SAM's inner objective's gradient ascent step involves the same gradient computation as standard SGD. Yet, when using standard SGD, we typically do not additionally smooth the loss function. For example, [4] recently proposed doing so, however, with the motivation of finding flatter minima (akin to SAM) and by sampling multiple perturbations instead of just one. If RS leads to more "stable" or "accurate" gradients, why not apply it to the descent step too? For example, in Algorithm 1, "Compute SAM gradient", why do we not additionally perturb $\mathbf w_{\text{adv}}$  as well if such smoothening is always beneficial?

Figure 1 shows Vanilla's cosine similarity between the standard and perturbed gradient decays faster than when RS gets applied. However, it is not clear to me why this matters and can't be simply resolved by re-scaling rho.

Figure 2 shows that the RS loss function increases faster in loss when rho is increased. Again, it is unclear why reaching higher losses by perturbing the weights further and further with increasing standard deviations is strictly better or important for ultimately reaching flatter (and therefore, better generalizing) minima.

The authors claim that Vanilla can not reach loss values as high as RS reaches (although this is not fully proven as rho's range ends at $\rho=1.0$). However, first perturbing the weights before approximating the sharpest loss in their neighborhood effectively moves the neighborhood ball in a worse-performing region -- it is not centered around $\mathbf w$ but at $\mathbf w + \mathbf \delta_0$. Since $\mathbf \delta_0 \neq \nabla L(\mathbf w)$ , almost surely $L(\mathbf w) < L(\mathbf w + \mathbf \delta_0)$ . Therefore, it is not surprising that an ascent step starting from $L(\mathbf w + \mathbf \delta_0)$ will reach even higher losses.

Further, we may note that the authors generated Figure 2 using the checkpoint of the 200th epoch on CIFAR-100. Using a decent learning rate/schedule, 200 epochs are enough for training a Resnet-18/WRN to (almost) convergence on CIFAR100. Hence, it is not clear why we want to reach higher-loss regions at this point during training.

Lastly, the loss function/y-axis of Figure 2 is not precisely specified. Assuming that it is sth like the cross-entropy loss and based on my experience, my guess would be that parameterizations with losses above 3 (which is what vanilla still reaches) resemble the performance of randomly initialized networks or within the first few optimization steps epochs during training. That is when the parameters are far from fruitful valleys and hence, not where we want the gradient ascent step jumping towards.

Despite these issues, I'd be easily convinced of RSAM's utility if the empirical results showed its superiority against other SAM variants and versatility across different tasks. Unfortunately, such results are largely missing.

* While the empirical performance gains over SAM are consistent across the shown experiments, they are less than 1% on average, which resembles the performance gains of other SAM variants, such as GSAM (cited in the paper), ASAM [1], SAM+SWA[3].  However, only for one task (model/dataset combination), the authors compare against GSAM, in which their performance differs by 0.1% and is identical (as the authors say themselves).
* I appreciate the additional R-GSAM experiment, but again, I find the number of tasks (which is only one here) too limited. It would be great to see how R-SAM compares against GSAM, ASAM, SWA+SAM and on more tasks, and even if they reach similar performances; it would then be interesting e.g. to combine these and see if they are complementary.
* This method introduces another hyper-parameter $\gamma$, which may incur additional tuning overhead for practictioners. While Figure 3c) shows that it is rather stable for three different values on one particular dataset, it would be nice to see a more thorough sensitivity analysis with e.g. more values or a default value used on a few more datasets/architectures (like SAM's rho=0.05).
* The paper includes thorough image classification results; however, it lacks empirical analyses on other tasks such as NLP or graph learning [2,3].
* All empirical results seem to be the result of one training run. It would great to see averaged results across at least three random seeds $\pm$ std.

Minor:
* Figures 1 and 2 show the effect of random smoothing (RS) on the loss surface. Since RS is a stochastic operation, it would be great to see some empirical confidence intervals (e.g., $\pm 1$ std across 3 seeds).



[1] ASAM: Adaptive Sharpness-Aware Minimization for Scale-Invariant Learning of Deep Neural Networks, Kwon et al, ICML 2021.
[2] Sharpness-Aware Minimization Improves Language Model Generalization, Bahri et al, ACL 2022.
[3] A Fair Comparison of Two Popular Flat-Minima Optimizers: Stochastic Weight Averaging vs. Sharpness-Aware Minimization, Kaddour et al, arXiv:2202.00661.
[4] Low-Pass Filtering SGD for Recovering Flat Optima in the Deep Learning Optimization Landscape, Bisla et al, 2022.

---

> ### Author Response · Authors · 2022-08-02
> **Response to Reviewer Dv1G**
>
> Thank you for your constructive and positive feedback, we carefully address your concerns below.
>
> ### **Q1:**
>
> > randomly smoothening (RS) the loss function significantly improves over a non-smoothed loss function for approximating the gradient ascent step -- is true. RS seems very heuristical to me; hence, I'd appreciate either more clarification on why it always improves (if so) or more discussion on when it may not hold and what its trade-offs are.
>
>
>
> Thanks for your comments. Let us restate the motivation of R-SAM.
>
>
> Consider the inner maximization problem of SAM within a region $\rho$. When the gradient is a constant within this region, then one-step gradient ascent can get the exact maximum. However, when the objective is very non-smooth -- which means gradient changes quickly within the region, one-step gradient ascent becomes a bad approximation. Therefore, when we smooth the inner maximization problem, the gradient will be more stable (change slower) within the region, and the one-step gradient ascent will obtain a better approximated solution for inner maximization. Based on that, in this paper, we try to firstly smooth the loss function and obtain the approximated solution on the smoothed loss function to simplify the above problem. We prove the smoothing effect in Theorem 1, and our algorithm is equivalent to conduct a gradient descent on the randomly smoothed function.
>
> The hyper-parameter $\gamma$ which controls the level of randomness is important. When $\gamma$ is small, the function is closer to the original one but the gradient is less smooth. When $\gamma$ is too large, the function will be very smooth but will be not close to the original function. For example, we try to provide the experimental results about the effects of $\gamma$ for the accuracy. We can find that it exists a trade-off and $\gamma$ is very important for the performance gain.
>
> |Model    |2e-3  | 1.5e-3  |1e-3  | 5e-4 | 1e-4 | 5e-5 |
> |---------|------|---------|------|------|------|------|
> |WRN-28-10| 84.7 | 85.1    | 85.3 | 85.4 | 85.1 | 84.5 |
> |ViT-B-16 | 79.7 | 80.2    | 80.7 | 80.7 | 80.4 | 79.7 |
> |ViT-S-16 | 77.3 | 77.9    | 78.5 | 78.2 | 77.9 | 77.8 |
>
> ### **Q2:**
>
> > SAM's inner objective's gradient ascent step involves the same gradient computation as standard SGD. Yet, when using standard SGD, we typically do not additionally smooth the loss function. For example, [4] recently proposed doing so, however, with the motivation of finding flatter minima (akin to SAM) and by sampling multiple perturbations instead of just one. If RS leads to more "stable" or "accurate" gradients, why not apply it to the descent step too? For example, in Algorithm 1, "Compute SAM gradient", why do we not additionally perturb $w_{adv}$ as well if such smoothening is always beneficial?
>
>
>
> Thanks for your comments. As you mentioned, [4] tries to use random noise to improve the performance in the gradient descent step. We think gradient ascent step and gradient descent step focus on different aspects. In the gradient ascent step, we do not need to consider the generalization problem and we will pay more attention to the solution of the maximization problem in Equation (1). However, for the gradient descent step, we do not only consider minimizing the loss value but also consider some other problems like the generalization problem.
>
> ### **Q3:**
>
> > Figure 1 shows Vanilla's cosine similarity between the standard and perturbed gradient decays faster than when RS gets applied. However, it is not clear to me why this matters and can't be simply resolved by re-scaling rho.
>
>
>
> Thanks for your comments. Vanilla SAM uses one-step gradient ascent to solve the inner maximization, which means that it uses the gradient $g(w)$ to approximate the gradient on the weight that locates in the trajectory from $w$ to $w_{adv} = w + \rho  \frac{g(w)}{||g(w)||}$. Gradient stability can be briefly defined as $\frac{||g(w) - g(w+\rho\frac{g(w)}{||g(w)||})||}{||\epsilon||}$, where $||\epsilon||$ is not 0. We use gradient similarity (cosine similarity between $g(w)$ and $g(w+\rho\frac{g(w)}{||g(w)||}))$ to approximate the definition when $\rho$ is fixed: $\frac{g(w) \cdot g(w+\rho\frac{g(w)}{||g(w)||})}{||g(w)|| ||g(w+\rho\frac{g(w)}{||g(w)||}||}$. Therefore, if the gradient is more stable in the trajectory from $w$ to $w_{adv}$, we can obtain a better approximated solution in the gradient ascent step.
>
>
>
> As for the reason why we can't be simply resolved by re-scaling is that we find that SAM is sensitive with the selection of $\rho$ value. Therefore, we would like to obtain a better solution when $\rho$ is fixed. If $\rho$ is too large, that may hurt the accuracy performance. In addition, the generalization will be hurt and it's difficult to converge to a flat region if $\rho$ is too low.

---

> > ### Author Response · Authors · 2022-08-02
> > **Response to Reviewer Dv1G - Continue**
> >
> > ### **Q4:**
> >
> > > Figure 2 shows that the RS loss function increases faster in loss when rho is increased. Again, it is unclear why reaching higher losses by perturbing the weights further and further with increasing standard deviations is strictly better or important for ultimately reaching flatter (and therefore, better generalizing) minima.
> >
> >
> >
> > Thanks for your comments. Increasing standard deviations can smooth the loss landscape based on Theorem 1. We would like to make sure $\frac{\alpha}{\gamma} < \beta$ to obtain smoother loss function. SAM uses one-step gradient ascent to solve the inner maximization problem to help the model converge to a flat region. we would like to use Figure 2 to show that random smoothing can obtain a larger loss value and obtain a better approximation for the inner maximization problem when $\rho$ is fixed. That may better help the model converge to a flat region.
> >
> >
> >
> > ### **Q5:**
> >
> > > The authors claim that Vanilla can not reach loss values as high as RS reaches (although this is not fully proven as rho's range ends at $\rho=1.0$). However, first perturbing the weights before approximating the sharpest loss in their neighborhood effectively moves the neighborhood ball in a worse-performing region -- it is not centered around $w$ but at $w+\delta_{0}$. Since $\delta_{0} \neq  \nabla L(w)$, almost surely $L(w) \leq L(w+\delta_{0})$. Therefore, it is not surprising that an ascent step starting from $L(w+\delta_{0})$ will reach even higher losses.
> >
> >
> >
> > Thanks for your comments. Vanilla SAM use one-step gradient ascent to approximate the solution of inner maximization. We find that one-step gradient ascent maybe difficult to solve the inner maximization problem when the model lies in a not very flat region. That motivates us to firstly smooth the loss landscape and then obtain the approximated solution in the smoothed loss function. We agree with you that random smoothing is easy to cause $L(w) \leq L(w+\delta_{0})$. However, our experimental results illustrate that the loss difference is minimal, especially compared with the difference between $L(w+\rho\frac{g(w)}{||g(w)||})$ and $L(w+\delta_{0}+\rho\frac{g(w)}{||g(w)||})$.
> >
> > ### **Q6:**
> >
> > > Further, we may note that the authors generated Figure 2 using the checkpoint of the 200th epoch on CIFAR-100. Using a decent learning rate/schedule, 200 epochs are enough for training a Resnet-18/WRN to (almost) convergence on CIFAR100. Hence, it is not clear why we want to reach higher-loss regions at this point during training.
> >
> >
> >
> > Thanks for your comments. The main reason that we use the checkpoint of 200th epoch is that related work [1] illustrates that the last epochs are more important for the performance gain. So we try to evaluate and analyze the smoothness at the last epoch.
> >
> > [1] Maksym Andriushchenko, Nicolas Flammarion. "Towards Understanding Sharpness-Aware Minimization". ICML 2022.
> >
> > ### **Q7:**
> >
> > > Lastly, the loss function/y-axis of Figure 2 is not precisely specified. Assuming that it is sth like the cross-entropy loss and based on my experience, my guess would be that parameterizations with losses above 3 (which is what vanilla still reaches) resemble the performance of randomly initialized networks or within the first few optimization steps epochs during training. That is when the parameters are far from fruitful valleys and hence, not where we want the gradient ascent step jumping towards.
> >
> >
> >
> > Thanks for your comments. The loss function in Figure 2 is cross-entropy loss.
> >
> > To obtain a better analysis and easy to understand the phenomenon, we try to expand the $\rho$ value.
> >
> > ### **Q8:**
> >
> > > While the empirical performance gains over SAM are consistent across the shown experiments, they are less than 1\% on average, which resembles the performance gains of other SAM variants, such as GSAM (cited in the paper), ASAM [1], SAM+SWA[3]. However, only for one task (model/dataset combination), the authors compare against GSAM, in which their performance differs by 0.1\% and is identical (as the authors say themselves).
> >
> >
> >
> >
> > Thanks for your comments. Our main contribution is to illustrate that random smoothing can also improve the performance and provide a new direction for SAM, which can make more people to further investigate it. We also try to combine our method with G-SAM for ViT on ImageNet and the improvement is about 0.4\%.

---

> > > ### Author Response · Authors · 2022-08-02
> > > **Response to Reviewer Dv1G - Continue**
> > >
> > > ### **Q9:**
> > >
> > > > I appreciate the additional R-GSAM experiment, but again, I find the number of tasks (which is only one here) too limited. It would be great to see how R-SAM compares against GSAM, ASAM, SWA+SAM and on more tasks, and even if they reach similar performances; it would then be interesting e.g. to combine these and see if they are complementary.
> > >
> > >
> > >
> > > Thanks for your comments. Due to the limited time, we provide the experimental results about R-GSAM on ViT in this table. We will report more results in the revision.
> > >
> > >  |Model   |AdamQ | SAM     | GSAM | R-SAM | R-GSAM|
> > > |---------|------|---------|------|-------|-------|
> > > |ViT-B-16 | 74.7 | 79.8    | 80.8 | 80.7  | 81.2  |
> > > |ViT-S-16 | 74.9 | 77.9    | 78.8 | 78.7  | 79.1  |
> > >
> > > From this table, we can find that R-SAM can achieve competitive accuracy compared with GSAM. In addition, R-GSAM can obtain better performance than GSAM.
> > >
> > > ### **Q10:**
> > >
> > > > This method introduces another hyper-parameter , which may incur additional tuning overhead for practictioners. While Figure 3c) shows that it is rather stable for three different values on one particular dataset, it would be nice to see a more thorough sensitivity analysis with e.g. more values or a default value used on a few more datasets/architectures (like SAM's rho=0.05).
> > >
> > >
> > >
> > >
> > > Thanks for your comments. We try to provide more thorough analysis about $\rho$ and $\lambda$ in the tables:
> > >
> > >  - Sensitivity Analysis about $\rho$
> > >
> > > |Model    | 0.8  | 1.0 | 1.2  | 1.3 | 1.4 | 1.5  | 1.6  | 1.8 | 2.0  |
> > > |---------|------|-----|------|-----|-----|------|------|-----|------|
> > > |ViT-B-16 | 80.2 | 80.3| 80.3 | 80.6| 80.7| 80.6 | 80.5 | 80.4| 80.1 |
> > >
> > >   - Sensitivity Analysis about $\lambda$
> > >
> > > |Model    | 0.5  | 1.0 | 1.5  | 2.0 | 2.5 | 3.0  |
> > > |---------|------|-----|------|-----|-----|------|
> > > |ViT-B-16 | 80.5 | 80.6 | 80.7| 80.7| 80.7| 80.7 |
> > >
> > > We can find that R-SAM shows a stable pattern for these parameters. We will update these results in the revision.
> > >
> > > ### **Q11:**
> > >
> > > > The paper includes thorough image classification results; however, it lacks empirical analyses on other tasks such as NLP or graph learning [2,3]. All empirical results seem to be the result of one training run. It would great to see averaged results across at least three random seeds std.
> > >
> > >
> > >
> > > Thanks for your comments. The reported experimental results are the mean accuracy with three random seed. We will also provide the error bar in the table:
> > >
> > > - CIFAR-10:
> > >
> > >
> > >
> > > |Model |SGD |SGD+M | RMSProp | AdamW | LPF-SGD |SAM | R-SAM |
> > > |---------|------|-------|---------|-------|---------|------|-------|
> > > |ResNet-18| 95.4 $\pm0.1$  | 95.6 $\pm0.2$  | 95.4 $\pm0.2$  | 95.1 $\pm0.1$  | 95.9 $\pm0.1$  | 96.4 $\pm0.2$  | 96.5 $\pm0.1$  |
> > > |ResNet-50| 95.8 $\pm0.1$  | 95.7 $\pm0.1$  | 95.7 $\pm0.1$  | 96.0 $\pm0.1$  | 96.3 $\pm0.2$  | 96.7 $\pm0.1$  | 96.9 $\pm0.1$  |
> > > |WRN-28-10| 96.4 $\pm0.1$  | 96.5 $\pm0.1$  | 96.4 $\pm0.2$ | 96.0 $\pm0.1$  | 96.8 $\pm0.1$ | 97.3 $\pm0.1$  | 97.5 $\pm0.1$ |
> > >
> > >
> > >
> > > - CIFAR-100
> > >
> > >
> > >
> > > |Model |SGD |SGD+M | RMSProp | AdamW | LPF-SGD |SAM | R-SAM |
> > > |---------|------|-------|---------|-------|---------|------|-------|
> > > |ResNet-18| 78.0 $\pm0.1$  | 78.9 $\pm0.2$  | 79.4 $\pm0.1$  | 77.7 $\pm0.1$ | 80.2 $\pm0.1$  | 80.9 $\pm0.2$  | 81.4 $\pm0.2$  |
> > > |ResNet-50| 80.9 $\pm0.2$ | 81.4 $\pm0.2$ | 81.4 $\pm0.1$  | 80.8 $\pm0.1$  | 82.1 $\pm0.2$  | 83.3 $\pm0.1$  | 84.0 $\pm0.1$ |
> > > |WRN-28-10| 81.1 $\pm0.1$  | 81.7 $\pm0.1$  | 81.7 $\pm0.1$ | 80.1 $\pm0.2$ | 82.6 $\pm0.1$ | 84.6 $\pm0.1$ | 85.2 $\pm0.1$  |
> > >
> > >
> > >
> > > - ImageNet
> > >
> > >
> > >
> > > |Model |AdamW | LPF-SGD |SAM | R-SAM |
> > > |---------|------|---------|------|-------|
> > > |ViT-B-16 | 74.7 $\pm0.1$  | 75.9 $\pm0.2$  | 79.8 $\pm0.1$ | 80.7 $\pm0.1$  |
> > > |ViT-S-16 | 74.9 $\pm0.2$  | 75.8 $\pm0.1$  | 77.9 $\pm0.2$  | 78.7 $\pm0.2$  |
> > >
> > >
> > >
> > >
> > > In addition, due to the limited time, we will further explore the proposed R-SAM in NLP and graph learning in our future work.

---

> > > > ### Author Response · Authors · 2022-08-02
> > > > **Response to Reviewer Dv1G - Continue**
> > > >
> > > > ### **Q12:**
> > > >
> > > > > It is not entirely clear why performing the inner maximization gradient ascent step on a smoothed loss function necessarily leads to a more "accurate" solution or exactly what that means.} Following the same argument, we would expect that we want to apply SGD (solving a minimization objective) on a smoothed loss function too. Yet, this is not standard practice. Works like [1] recently proposed to explore such smoothening; however, with the motivation of finding flat minima (akin to SAM), not "gradient stability" or more "accurate" solutions. It would be great if the authors could clarify why "gradient stability" is important for inner maximization.
> > > >
> > > >
> > > >
> > > > Thanks for your comments.  As shown in the previous answer about Q1,  let us restate the motivation of R-SAM.
> > > >
> > > > Consider the inner maximization problem of SAM within a region  \rhoρ. When the gradient is a constant within this region, then one-step gradient ascent can get the exact maximum. However, when the objective is very non-smooth – which means gradient changes quickly within the region, one-step gradient ascent becomes a bad approximation. Therefore, when we smooth the inner maximization problem, the gradient will be more stable (change slower) within the region, and the one-step gradient ascent will obtain a better approximated solution for inner maximization. Based on that, in this paper, we try to firstly smooth the loss function and obtain the approximated solution on the smoothed loss function to simplify the above problem. We prove the smoothing effect in Theorem 1, and our algorithm is equivalent to conduct a gradient descent on the randomly smoothed function.
> > > >
> > > > The hyper-parameter $\gamma$ which controls the level of randomness is important. When $\gamma$ is small, the function is closer to the original one but the gradient is less smooth. When $\gamma$ is too large, the function will be very smooth but will be not close to the original function.

---

> > > > > ### Comment · Reviewer_Dv1G · 2022-08-04
> > > > > **Thanks for the detailed responses!**
> > > > >
> > > > > Thank you for your detailed responses.
> > > > >
> > > > > My concerns have been mostly addressed, and I raised my score.

---

### Official Review · Reviewer_tXqv · 2022-07-10

**Rating:** 6
**Confidence:** 4
**Soundness:** 3 good
**Presentation:** 3 good
**Contribution:** 3 good

**Summary:**

This work analyzes the approximation quality of the inner maximization term in Sharpness-Aware Minimization (SAM). It is shown that the unstable gradients can hurt the inner maximization. A novel method called Random Sharpness-Aware Minimization (R-SAM) is proposed that smooths the landscape, enabling a better approximation of the inner maximization. This is done by the addition of Gaussian Noise before the computation of the inner maximization term. The results are shown on CIFAR-10, CIFAR-100, and ImageNet datasets. R-SAM improves upon the performance of SAM in most settings.

**Questions:**

1. In Figures 1 and 2, it would be clearer if the authors also mentioned the model's accuracy in a Table. For instance, $\rho$ = 1.0 leads to the max loss value in Figure 2. Is that observed in the performance as well?
(Also, why is the scale of the rho value in both the figures very different?)

2. The experimental details of the CIFAR-100 experiments, including the $\rho$ value, seem to be missing. A note comparing the $\rho$ value with SAM and R-SAM can be added. It is to be observed whether training with high $\rho$ is also stable for ResNets. With ViTs (Table 6), the optimal $\rho$ value is higher for R-SAM.

3. The details regarding the value of $\gamma$ have not been mentioned. A sensitivity analysis of the value of $\gamma$ can also be added.

4. Question 3c of the checklist is answered as, "It would be too computationally expensive for us. We conduct all the experiments at least three times and report the average value." This is a bit confusing. I just wanted to clarify whether the authors have run all the experiments three times. This is significant because the increase with R-SAM is small in many experiments and within the error bound (as mentioned in SAM).

4. In the results section, it is unclear why the authors are discussing the improvement of the original SAM over SGD+M and AdamW (L233-238, L246-250, L261-264). These results have been discussed in the SAM work.

5. The method proposed by Chaudhari et al. is called Entropy-SGD and not "Energy-SGD," as mentioned by the authors. (L53)

**Limitations:**

The limitations of the work is not mentioned. This work has no potential negative societal impact.

**Strengths And Weaknesses:**

Strengths:
1. The analysis of the effect of Gaussian noise on the perturbed loss-value (Figure 2) is interesting. The analysis of approximation of gradient in SAM has not been studied and is a good direction.
2. The proposed method, R-SAM, is simple, has no additional cost compared to SAM, and allows for stable training with large $\rho$.
3. The results on combining G-SAM with R-SAM, resulting in R-GSAM, show that the R-SAM can be integrated with other variants of SAM.

Weaknesses:
1. Relationship between stable gradient and approximation of inner maximization term is not very clear. Also, the term stable gradient is not well-defined. If a stable gradient were the goal, then $\rho$ = 0 (SGD) or a low $\rho$ would lead to the most stable gradient according to the cosine definition. However, it is observed that a very low $\rho$ value does not lead to a significant gain in accuracy over SGD.
3. LPF-SGD [1] is a recently proposed method that aims to reach flat optima. The algorithm of LPF-SGD also adds Gaussian Noise to the parameters with the variance equal to the norm of filter weights, similar to R-SAM.
4. The increase in accuracy of R-SAM in comparison to SAM is minimal in most settings (Table 1, Table 2, Figure 3a).

[1] Devansh Bisla, Jing Wang, and Anna Choromanska. Low-pass filtering sgd for recovering flat optima in the deep learning optimization landscape. arXiv preprint arXiv:2201.08025, 2022.

---

> ### Author Response · Authors · 2022-08-02
> **Response to Reviewer tXqv**
>
>   Thanks for your insightful comments, we carefully address your concerns below.
>
> ### **Q1:**
>
> > Relationship between stable gradient and approximation of inner maximization term is not very clear. Also, the term stable gradient is not well-defined. If a stable gradient were the goal, then $\rho = 0$ (SGD) or a low $\rho$ would lead to the most stable gradient according to the cosine definition. However, it is observed that a very low $\rho$ value does not lead to a significant gain in accuracy over SGD.
>
> Thanks for your comments. By gradient stability, we mean the Lipchitz constant of gradient, i.e., $\frac{||g(w)-g(w+\epsilon)||}{||\epsilon||}$, which measures how a perturbation on $w$ will change the gradient. When this value equals to $0$ within distance $\rho$, the gradient is a constant within the region so one-step gradient ascent can get the maximum. However, when the value is large (gradient is unstable), since the gradient at $w$ can be very different from other points within the region, it is harder to solve the inner maximization problem. Computing this constant over the whole region is difficult, so we use $||g(w) - g(w+\rho\frac{g(w)}{||g(w)||})||/\rho$ as the proxy.
>
> In this paper, we discuss gradient stability in the context of a fixed $\rho$ (SAM's parameter) so it is related to how good can we solve it with one-step gradient ascent, as mentioned above. When $\rho$ is small, we can solve the inner maximization problem better but also SAM will encounter worse performance.
>
> ### **Q2:**
>
> > LPF-SGD [1] is a recently proposed method that aims to reach flat optima. The algorithm of LPF-SGD also adds Gaussian Noise to the parameters with the variance equal to the norm of filter weights, similar to R-SAM.
>
>
>
> Thanks for your comments. LPF-SGD try to use gaussian noise to directly perturb the weight in the gradient descent step. However, we try to use gaussian noise to smooth the loss landscape in the gradient ascent step. We also try to compare our proposed R-SAM with LPF-SGD and the experimental result is shown in the Table 1 and Table 2 of the revision.
>
>   - CIFAR-100
>
> |Model |SGD |SGD+M | RMSProp | AdamW | LPF-SGD |SAM | R-SAM |
> |---------|------|-------|---------|-------|---------|------|-------|
> |ResNet-18| 78.0 | 78.9 | 79.4 | 77.7 | 80.2 | 80.9 | 81.4 |
> |ResNet-50| 80.9 | 81.4 | 81.4 | 80.8 | 82.1 | 83.3 | 84.0 |
> |WRN-28-10| 81.1 | 81.7 | 81.7 | 80.1 | 82.6 | 84.6 | 85.2 |
>
>
>   - ImageNet
>
> |Model |AdamW | LPF-SGD |SAM | R-SAM |
> |---------|------|---------|------|-------|
> |ViT-B-16 | 74.7 | 75.9 | 79.8 | 80.7 |
> |ViT-S-16 | 74.9 | 75.8 | 77.9 | 78.7 |
>
> ### **Q3:**
>
> > The increase in accuracy of R-SAM in comparison to SAM is minimal in most settings (Table 1, Table 2, Figure 3a).
>
>
>
> Thanks for your comments. SAM has achieved a great accuracy (about 97\%) on CIFAR-10. So there is relatively small room for improvements. On CIFAR-100, SAM already achieve great performance but we noticed that R-SAM can consistently achieve about 0.5\% accuracy improvement over SAM.
>
> Chen et al. illustrates that SAM can obtain a great performance on ViT and therefore our main experimental results focus on ViT (Table 3, Table 4 and Figure 3). We can find that R-SAM can achieve about 0.9\% improvement for ViT-B-16.
>
>
>
> [1] Chen, Xiangning, Cho-Jui Hsieh, and Boqing Gong. 'When vision transformers outperform ResNets without pre-training or strong data augmentations.' ICLR 2022.
>
> ### **Q4:**
>
> > In Figures 1 and 2, it would be clearer if the authors also mentioned the model's accuracy in a Table. For instance, $\rho$ = 1.0 leads to the max loss value in Figure 2. Is that observed in the performance as well? (Also, why is the scale of the rho value in both the figures very different?)
>
>
>
> Sorry for the confusion. It is not true that a larger value in Figure 2 implies better performance. In SAM's paper, it has been shown that larger $\rho$ will over-regularize the model which leads to degraded prediction accuracy. Therefore, for larger $\rho$ such as $\rho=1$, even if we can solve the inner maximization very well, it is not helpful for getting a better model.
>
>
>
> Due to the limited time, we mainly focus on the model's accuracy when $\rho=0.1, 0.2$ and $0.3$. We will provide more experimental results about Figure 2 in the revision.
>
>
>
> |Model |$\rho=0.1$ |$\rho=0.2$ |$\rho=0.3$ |
> |---------|-----------|-----------|-----------|
> |WRN-28-10| 84.3 | 84.9 | 85.2 |

---

> > ### Author Response · Authors · 2022-08-02
> > **Response to Reviewer tXqv - Continue**
> >
> > ### **Q5:**
> >
> > > The experimental details of the CIFAR-100 experiments, including the $\rho$ value, seem to be missing. A note comparing the value with SAM and R-SAM can be added. It is to be observed whether training with high $\rho$ is also stable for ResNets. With ViTs (Table 6), the optimal value is higher for R-SAM.
> >
> >
> >
> > Thanks for your comments. Let me briefly introduce the experimental details of CIFAR-100 training.
> >
> > |Model            |Batch Size   |Epoch  | LR      | Weight Decay | $\rho$ |$\lambda$|
> > |-----------------|-------------|-------|---------|--------------|---------|------|
> > |WRN-28-10 + SAM  | 128         | 200   | 0.1     | 5E-3         | 0.1     | /    |
> > |WRN-28-10 + R-SAM| 128         | 200   | 0.1     | 5E-3         | 0.3     | 1 |
> > |ResNet-50 + SAM  | 128         | 200   | 0.1     | 5E-3         | 0.1     | /    |
> > |ResNet-50 + R-SAM| 128         | 200   | 0.1     | 5E-3         | 0.3     | 1 |
> >
> >
> >
> > We will provide more details in the supplement.
> >
> > ### **Q6:**
> >
> > > The details regarding the value of $\gamma$ have not been mentioned. A sensitivity analysis of the value of $\gamma$ can also be added.
> >
> >
> >
> > Thanks for your comments. We will provide the analysis about $\gamma$ on ViT training in the revision paper. We can briefly introduce the results:
> >
> >
> >
> > |Model |2e-3 | 1.5e-3 |1e-3 | 5e-4 |
> > |---------|------|---------|------|-------|
> > |WRN-28-10| 84.7 | 85.1 | 85.3 | 85.4 |
> > |ViT-B-16 | 79.7 | 80.2 | 80.7 | 80.7 |
> > |ViT-S-16 | 77.3 | 77.9 | 78.5 | 78.2 |
> >
> >
> >
> > We can find that $\gamma$ shows a relatively stable pattern. For most cases, R-SAM can achieve a better performance compared with vanilla SAM.
> >
> > ### **Q7:**
> >
> > > Question 3c of the checklist is answered as, "It would be too computationally expensive for us. We conduct all the experiments at least three times and report the average value." This is a bit confusing. I just wanted to clarify whether the authors have run all the experiments three times. This is significant because the increase with R-SAM is small in many experiments and within the error bound (as mentioned in SAM).
> >
> >
> >
> > Thanks for your comments. We have run the experiments for 3 times and report the mean value in the tables of the paper. We will update it in the revision and also provide the error bar in the table:
> >
> >
> >
> > - CIFAR-10:
> >
> >
> >
> > |Model |SGD |SGD+M | RMSProp | AdamW | LPF-SGD |SAM | R-SAM |
> > |---------|------|-------|---------|-------|---------|------|-------|
> > |ResNet-18| 95.4 $\pm0.1$  | 95.6 $\pm0.2$  | 95.4 $\pm0.2$  | 95.1 $\pm0.1$  | 95.9 $\pm0.1$  | 96.4 $\pm0.2$  | 96.5 $\pm0.1$  |
> > |ResNet-50| 95.8 $\pm0.1$  | 95.7 $\pm0.1$  | 95.7 $\pm0.1$  | 96.0 $\pm0.1$  | 96.3 $\pm0.2$  | 96.7 $\pm0.1$  | 96.9 $\pm0.1$  |
> > |WRN-28-10| 96.4 $\pm0.1$  | 96.5 $\pm0.1$  | 96.4 $\pm0.2$  | 96.0 $\pm0.1$  | 96.8 $\pm0.1$  | 97.3 $\pm0.1$  | 97.5 $\pm0.1$  |
> >
> >
> >
> > - CIFAR-100
> >
> >
> >
> > |Model |SGD |SGD+M | RMSProp | AdamW | LPF-SGD |SAM | R-SAM |
> > |---------|------|-------|---------|-------|---------|------|-------|
> > |ResNet-18| 78.0 $\pm0.1$  | 78.9 $\pm0.2$  | 79.4 $\pm0.1$  | 77.7 $\pm0.1$  | 80.2 $\pm0.1$ | 80.9 $\pm0.2$ | 81.4 $\pm0.2$  |
> > |ResNet-50| 80.9 $\pm0.2$  | 81.4 $\pm0.2$  | 81.4 $\pm0.1$  | 80.8 $\pm0.1$  | 82.1 $\pm0.2$  | 83.3 $\pm0.1$  | 84.0 $\pm0.1$  |
> > |WRN-28-10| 81.1 $\pm0.1$  | 81.7 $\pm0.1$  | 81.7 $\pm0.1$  | 80.1 $\pm0.2$  | 82.6 $\pm0.1$  | 84.6 $\pm0.1$  | 85.2 $\pm0.1$  |
> >
> >
> >
> > - ImageNet
> >
> >
> >
> > |Model |AdamW | LPF-SGD |SAM | R-SAM |
> > |---------|------|---------|------|-------|
> > |ViT-B-16 | 74.7 $\pm0.1$  | 75.9 $\pm0.2$  | 79.8 $\pm0.1$  | 80.7 $\pm0.1$  |
> > |ViT-S-16 | 74.9 $\pm0.2$  | 75.8 $\pm0.1$  | 77.9 $\pm0.2$  | 78.7 $\pm0.2$  |
> >
> > ### **Q8:**
> >
> > > In the results section, it is unclear why the authors are discussing the improvement of the original SAM over SGD+M and AdamW (L233-238, L246-250, L261-264). These results have been discussed in the SAM work.
> >
> >
> >
> > Thanks for your comments. We will rewrite the part in the experiments.
> >
> > ### **Q9:**
> >
> > > The method proposed by Chaudhari et al. is called Entropy-SGD and not "Energy-SGD," as mentioned by the authors. (L53)
> >
> >
> >
> > Thanks for your comments. We have corrected it in the revision.

---

> > > ### Comment · Reviewer_tXqv · 2022-08-06
> > > **Response to Author**
> > >
> > > I thank the authors for a detailed response. The additional experimental analysis including the comparison with LPF-SGD, Hessian Eigenvalue analysis (Q5 of uTRr) and the sensitivity analysis of $\gamma$ strengthen the paper. I have increased my score accordingly.

---

### Official Review · Reviewer_uTRr · 2022-07-11

**Rating:** 8
**Confidence:** 3
**Soundness:** 3 good
**Presentation:** 4 excellent
**Contribution:** 3 good

**Summary:**

Sharpness-Aware Minimization (SAM) is an emerging deep learning technology in recent years, which aims to find parameters lie in a "flat" region of the energy landscape. Its purpose can be formulated as a minimax problem as shown in (4): A parameter vector w_adv is "optimal" in the sense of SAM if the loss of worst solution in the neighborhood of w_adv is minimized. The vanilla SAM solved this problem by approximating the landscape around a solution w as a linear function. This paper investigates this approximation thoroughly by experimental and theoretical analysis, and shows that this approximation is possibly poor in applications. Inspired by recent works in the field of adversarial training, authors improve the vanilla by introducing noise wo weight parameters.


**Questions:**

1) In Fig. 1, the "flatness" of w is measured by the cosine value between g(w) and g(w+rho*g(w)/||g(w)||). But is it a suitable measure for all w? If w is close to a local minimum and g(w) is close to zero, then directions of both g(w) and g(w+rho*g(w)/||g(w)||) are "random", and the cosine value could be small even in a very flat region. The detailed settings of experiments in Fig. 1 are not provided. If the w is obtained after training, we need to consider this case.

2) There are some typos in figures. In Fig. 1, \hat g(w) should be g(\hat w). In Fig. 2, L(w+rho*g(w)/g(w)) should be g(w+rho*g(w)/||g(w)||).

3) Do authors miss the mathematical expectation notation somewhere? For example, in Theorem 1, L_s(w)=L(w+delta_0) or E[L(w+delta_0)]? (If it is the former one, we cannot say that L_s(w) must have a smaller Lipschitz constant.)

4) Baseline methods include SGD (and modified SGD) without SAM and existing SAM methods. Certainly it is impossible to compare the proposed method with all regularization techniques, but it would be better to consider some advanced non-SAM strategies here.

5) As stated by authors, "deep learning is usually sensitive to hyper-parameters". So the small improvement of the accuracy cannot strongly prove that RSAM significantly outperforms the other SAM methods. I suggest to add some numerical results to show that RSAM can find "flatter areas", which is more meaningful than 0.1% improvement in accuracy.

**Limitations:**

Authors have adequately addressed the limitations and potential negative societal impact of their work.

**Strengths And Weaknesses:**

Strengths:

Both theoretical and numerical analysis is thorough. The idea is simple and easy-to-use. The paper is clearly written and the contribution is significant.

Weakness:

See Questions.

---

> ### Author Response · Authors · 2022-08-02
> **Response to Reviewer uTRr**
>
> Thanks for your constructive and positive feedback, we carefully address your concerns below.
>
> ### **Q1:**
>
> > In Fig. 1, the "flatness" of w is measured by the cosine value between g(w) and $g(w+ \rho g(w)/||g(w)||$). But is it a suitable measure for all w? If w is close to a local minimum and g(w) is close to zero, then directions of both g(w) and $g(w+ \rho g(w)/||g(w)||)$ are "random", and the cosine value could be small even in a very flat region. The detailed settings of experiments in Fig. 1 are not provided. If the w is obtained after training, we need to consider this case.
>
>
>
> Sorry for the confusion. We use the weight $w$ after training for Figure 1. For this experiment, we observe the gradient is not very small during the training process. For example, we find that the gradient norm varies between 2 and 4.
>
> ### **Q2:**
>
> > There are some typos in figures. In Fig. 1, $\hat g(w)$ should be $g(\hat w)$. In Fig. 2, $L(w+\rho g(w)/g(w))$ should be $g(w+\rho g(w)/||g(w)||)$.
>
>
>
> Sorry for the confusion. We have corrected them in the revision.
>
> ### **Q3:**
>
> > Do authors miss the mathematical expectation notation somewhere? For example, in Theorem 1, $L_s(w)=L(w+\delta_0)$ or $E[L(w+\delta_0)]$? (If it is the former one, we cannot say that $L_s(w)$ must have a smaller Lipschitz constant.)
>
>
> Sorry for the confusion. We have clarified the description in the revision. For Theorem 1, $L_{S}(\boldsymbol{ w}) = L(\boldsymbol{ w} + \boldsymbol{ \delta_{0}})$ is more smooth than the original loss function $L(\boldsymbol{w})$ when $\frac{\alpha}{\gamma} \leq  \beta$.
>
> ### **Q4:**
>
> > Baseline methods include SGD (and modified SGD) without SAM and existing SAM methods. Certainly it is impossible to compare the proposed method with all regularization techniques, but it would be better to consider some advanced non-SAM strategies here.
>
>
>
> Thanks for your comments. We try to use weight decay, label smoothing, dropout and data augmentation for the experiments in Table 3 and Table 4. We have provided these details in the revision. In addition, we try to provide more experimental results about non-SAM method. Due to the limited time, we firstly try to compare our proposed method with LPF-SGD in the revision.
>
>
>   - CIFAR-100
>
> |Model |SGD |SGD+M | RMSProp | AdamW | LPF-SGD |SAM | R-SAM |
> |---------|------|-------|---------|-------|---------|------|-------|
> |ResNet-18| 78.0 | 78.9 | 79.4 | 77.7 | 80.2 | 80.9 | 81.4 |
> |ResNet-50| 80.9 | 81.4 | 81.4 | 80.8 | 82.1 | 83.3 | 84.0 |
> |WRN-28-10| 81.1 | 81.7 | 81.7 | 80.1 | 82.6 | 84.6 | 85.2 |
>
>
>   - ImageNet
>
> |Model |AdamW | LPF-SGD |SAM | R-SAM |
> |---------|------|---------|------|-------|
> |ViT-B-16 | 74.7 | 75.9 | 79.8 | 80.7 |
> |ViT-S-16 | 74.9 | 75.8 | 77.9 | 78.7 |
>
>
>
> From the above experimental results, we can find that although the non-sam method LPF-SGD can achieve a great performance compared with traditional optimizers (SGD+M, AdamW), It still exists a performance gap between LPF-SGD and SAM.
>
> ### **Q5:**
>
> > As stated by authors, "deep learning is usually sensitive to hyper-parameters". So the small improvement of the accuracy cannot strongly prove that RSAM significantly outperforms the other SAM methods. I suggest to add some numerical results to show that RSAM can find "flatter areas", which is more meaningful than 0.1% improvement in accuracy.
>
>
>
> Thanks for your constructive comments. We try to use Hessian dominate eigenvalue to analyze the flatter areas that SAM and R-SAM learn in ViT. Due to the limited time, we mainly focus on the analysis about ViT-B-16. We will provide more analysis in the revision. The result is shown in the table:
>
> |Model |AdamW |SAM | R-SAM |
> |---------|------|------|-------|
> |ViT-B-16 | 727.1 | 21.2 | 17.4 |
> |ViT-S-16 | 571.2 | 20.7 | 11.1 |
>
> From this table, we can find that SAM means a lower eigenvalue of Hessian and therefore promote the model converge to a flat region. In addition, R-SAM can further reduce the eigenvalue of Hessian and converge to flatter region.  We will explore it further in the future.
>
> [1] Xiangning Chen, Cho-Jui Hsieh, Boqing Gong. "When Vision Transformers Outperform ResNets without Pre-training or Strong Data Augmentations" ICLR 2022.

---

### Official Review · Reviewer_ciRx · 2022-07-11

**Rating:** 4
**Confidence:** 5
**Soundness:** 3 good
**Presentation:** 2 fair
**Contribution:** 2 fair

**Summary:**

The paper proposes to modify the Sharpness-Aware Minimization (SAM) algorithm to include a random perturbation before taking the gradient ascept step. The paper presents some justifications to why it could be beneficial, based on the smoothing argument. The experimental results show the effectiveness of the proposed method.


**Questions:**

- Was a grid search done for SAM (e.g., Table 1 and 2)? And how are the hyperparameters tuned for each dataset for the proposed method, R-SAM?


**Limitations:**

The proposed method has three hyperparameters: the perturbation radius $\rho$, $\gamma$ for the noise standard deviation of the noise step, and $\lambda$ for the step size of the gradient. In comparison, SAM has only one hyperparameter. The hyperparameter sensitivity presented in **4.6 Sensitivity Analysis** is not very convincing as it shows only one-dimensional grid search (while keeping the other two dimensions fixed to the optimal values) and, for example, the range of $\rho$ presented in Fig. 3 is in fact quite narrow (from 1.2 to 1.6).

---

**Update after the rebuttal**
*Overall, I'd say that the improvements coming from R-SAM are consistent across multiple settings which makes the proposed method practically useful. However, the core idea of simply adding random noise (although with specific covariance) before the ascent step of SAM is extremely simple and lacks novelty. Moreover, I think the paper could've been implemented more carefully in various aspects (mentioned above) such as careful ablation studies and better motivation of the proposed method. I strongly feel that since the proposed method is so simple, the paper has to compensate it with very careful and rigorous empirical validations. I increase my score to 4 in light of the improvements of the paper presented during the rebuttal but still feel that the paper is below the acceptance bar.*

**Strengths And Weaknesses:**

- **Originality.** Low. The idea of adding noise before taking the gradient ascent step of SAM is very simple.
- **Quality.** Medium. The experimental evaluation of R-SAM is quite convincing. The findings that SAM/R-SAM can quite noticeably boost the OOD performance (ImageNet-R, ImageNet-C, etc) is also nice. On the other hand, the motivation of the proposed method requires significant improvements (see below).
- **Clarity.** Medium. Most of the content is clear.
- **Significance.** Overall, I appreciate the empirical effectiveness of the proposed method which could be of interest to practitioners. However, the paper requires multiple improvements listed below.

Things that need improvement:
- A crucial baseline, Bisla et al. (AISTATS’22), is cited but isn’t compared to or discussed in sufficient depth. A comparison to their method (which is **not** using worst-case perturbations) would be beneficial, especially since they leverage random perturbations with the same covariance as suggested in this paper. I wonder if it’s key to the performance of this method.
- *“That means the gradient of smoothed weight $w + \delta_0$ can better maximize the loss value in a large region.”* – But do we really need such large weight perturbations? For $\rho=1.0$, the training loss is increased to $\approx 6$ (Fig. 2) which is even higher than the loss of a random classifier ($\approx ln(100) \approx 4.6$).
- Fig. 1 and 2: is it done for test-time BatchNorm? If yes, then it should be remade for training-time BatchNorm or for a network trained with a different normalization method. In practice, the difference between sharpness estimates of test-time and training-time BatchNorm is very significant (see, e.g., https://arxiv.org/abs/2206.06232).
- The loss values in Fig. 2 are checked for the algorithm described in Eq. (6) while the proposed algorithm of R-SAM presented in Eq. (7) differs from Eq. (6) by the presence of the projection and a special covariance matrix of the noise. In order to properly motivate R-SAM (and not some other possible variation), one has to perform the experiment in Fig. 2 specifically for R-SAM. Intuitively, the projection step can make a lot of difference and I’m not sure if the algorithm in Eq. (7) will lead to a more accurate maximizer compared to the vanilla SAM.
- In addition, the role of the noise covariance is not sufficiently discussed. It’s unclear whether it’s crucial or not. For such a simple proposed idea, I’d expect much more rigorous ablation study for the key elements of the proposed method.
- The original SAM paper already pointed out that multiple steps of projected gradient ascent aren’t helpful (see Table 11 in https://arxiv.org/abs/2010.01412). That would be a more direct way to more accurately solve the inner maximization.
- Error bars are necessary to claim that the observed improvement is indeed significant.


In multiple places throughout the paper, the writing is not precise and some claims don’t seem to be justified:
- “However, the model usually locates in the sharp minima [31] where the unstable gradient, to a large extent, makes one-step gradient ascent performs poorly.” – This statement is surprising given that SAM aims to minimize sharpness and converge to a flat region. Moreover, this sentence leads to the impression that [31] demonstrated that this is somehow the case, although [31] has nothing to do with this.
- “Therefore, how to improve the approximation of the inner maximization procedure and obtain a more aggressive weight is an important problem” – What is meant by the “aggressive weight”?
- “However, the loss landscape of neural network is usually sharp and non-linear [30, 34].” – Surprising to see statements like this as the loss landscape of a **linear** model is also non-linear and may be considered sharp depending on the data covariance.
- “we observe that models can still locate in a sharp region, leading to poor performance on one-step gradient ascent.” – This statement isn’t complemented by sharpness measurements.
- “we are able to apply the one-step gradient ascent on the smoothed weights for a much more accurate measurement of the inner maximization” – There is no evidence to this in the paper. Fig. 2 is done for a different algorithm (see my concern about Eq. (6) vs. Eq. (7)).
- Eq. (2): there should be a division by the gradient norm, not multiplication.

---

> ### Author Response · Authors · 2022-08-02
> **Response to Reviewer ciRx**
>
> Thanks for your constructive feedback, we carefully address your concerns below.
>
>
>
> ### **Improvement 1:**
>
> > A crucial baseline, Bisla et al. (AISTATS’22), is cited but isn’t compared to or discussed in sufficient depth. A comparison to their method (which is not using worst-case perturbations) would be beneficial, especially since they leverage random perturbations with the same covariance as suggested in this paper. I wonder if it’s key to the performance of this method.
>
>
>
> Thanks for your constructive suggestion. We will provide more discussion about LPF-SGD in the revision. More specially, we try to compare R-SAM with LPF-SGD and the experimental results are shown in the following tables:
>
> - CIFAR-100
>
> |Model |SGD |SGD+M | RMSProp | AdamW | LPF-SGD |SAM | R-SAM |
> |---------|------|-------|---------|-------|---------|------|-------|
> |ResNet-18| 78.0 | 78.9 | 79.4 | 77.7 | 80.2 | 80.9 | 81.4 |
> |ResNet-50| 80.9 | 81.4 | 81.4 | 80.8 | 82.1 | 83.3 | 84.0 |
> |WRN-28-10| 81.1 | 81.7 | 81.7 | 80.1 | 82.6 | 84.6 | 85.2 |
>
>
>   - ImageNet
>
> |Model |AdamW | LPF-SGD |SAM | R-SAM |
> |---------|------|---------|------|-------|
> |ViT-B-16 | 74.7 | 75.9 | 79.8 | 80.7 |
> |ViT-S-16 | 74.9 | 75.8 | 77.9 | 78.7 |
>
>
>
> From these tables we can find that our method is better than LPF-SGD, so integrating randomness in inner gradient ascent is important to achieve better performance.
>
>
>
> ### **Improvement 2**：
>
> > 'That means the gradient of smoothed weight can better maximize the loss value in a large region.' – But do we really need such large weight perturbations? For $\rho$=1.0, the training loss is increased to $\approx$ 6.0 (Fig. 2) which is even higher than the loss of a random classifier ($\approx$ ln(100) $\approx$ 4.6).
>
>
>
> Thanks for your comments. We have modified this in the revision. In order to demonstrate the empirical intuition more clearly, we use a larger $\rho$ value to illustrate the long-term trend. Actually, from the figures, we can find that a stronger Gaussian perturbation as the initialization can obtain a larger loss value for the inner maximization when $\rho$ is 0.3 for most cases, which is close to the $\rho$ values we used in practice.
>
>
>
> ### **Improvement 3:**
>
> > Improvement: Fig. 1 and 2: is it done for test-time BatchNorm? If yes, then it should be remade for training-time BatchNorm or for a network trained with a different normalization method. In practice, the difference between sharpness estimates of test-time and training-time BatchNorm is very significant (see, e.g., https://arxiv.org/abs/2206.06232)
>
>
> Thanks for your comments and reference. We use training-time BatchNorm for Figure 1 and Figure 2 in the original submission. Based on the conclusion in your provided reference, we have clarified it in the revision.
>
> ### **Improvement 4:**
>
> > The loss values in Fig. 2 are checked for the algorithm described in Eq. (6) while the proposed algorithm of R-SAM presented in Eq. (7) differs from Eq. (6) by the presence of the projection and a special covariance matrix of the noise. Intuitively, the projection step can make a lot of difference and I’m not sure if the algorithm in Eq. (7) will lead to a more accurate maximizer compared to the vanilla SAM.
>
>
>
> Equation (6):
>
>   $\boldsymbol{w_{adv}} = \boldsymbol{\hat{w}} + \rho_{1} \frac{\boldsymbol{g(\hat{w})}}{\|\boldsymbol{g(\hat{w})}\|} = \boldsymbol{w} + \boldsymbol{\delta_{0}} + \rho_{1} \frac{\boldsymbol{g(w+\delta_{0})}}{\|\boldsymbol{g(w+\delta_{0})}\|}$
>
>
>
> Equation (7):
>
>
>
> $\boldsymbol{w_{adv}} = \boldsymbol{w} + \rho_{2} \frac{\boldsymbol{\delta_{0}}+\lambda\boldsymbol{g(\hat{w})}}{||\boldsymbol{\delta_{0}} + \lambda  \boldsymbol{g(\hat{w})}||}$
>
>
>
> Thanks for your comments. In fact, we can show that Equation (6) and Equation (7) share the similar form, except with a difference scaling constant.
>
>
>
> Firstly, we try to analyze Equation (7):
>
>
>
> $\boldsymbol{w_{adv}} = \boldsymbol{w} + \rho_{2} \frac{\boldsymbol{\delta_{0}}+\lambda  \boldsymbol{g(\hat{w})}}{||\boldsymbol{\delta_{0}} + \lambda  \boldsymbol{g(\hat{w})}||}
> = \boldsymbol{w} + \rho_{2} \frac{\boldsymbol{\delta_{0}}}{\|\boldsymbol{\delta_{0}} + \lambda  \boldsymbol{g(\hat{w})}\|} + \rho_{2} \frac{\lambda  \boldsymbol{g(\hat{w})}}{\|\boldsymbol{\delta_0}+\lambda\boldsymbol{g(\hat{w})}\|}$
>
>
>
> If $\rho_{2} = \|\delta_{0}+\lambda g(\hat{w})\|$ and $\lambda = \frac{\rho_{1}}{\|g(w+\delta_{0})\|}$, we can find that Equatuon (6) is equal to Equation (7). If not, we can tune the value of $\rho_{2}$ and $\lambda$ to make sure that $\rho_{2} \frac{\lambda}{\|\delta_{0} + \lambda g(\hat{w})\|} = \frac{\rho_{1}}{\|g(w+\delta_{0})}$. Based on that, the main difference between Equation (6) and Equation (7) will be the term about random noise $\delta_{0}$: $\delta_{0}$ in Equation (6) and $\frac{\rho_{2}}{\|\delta_{0} + \lambda g(\hat{w})\|}\delta_{0}$. Therefore, Equation (7) can be seen as introducing additional random noise compared with Equation (6).

---

> > ### Author Response · Authors · 2022-08-02
> > **Response to Reviewer ciRx - Continue**
> >
> > ### **Improvement 5:**
> >
> > > In addition, the role of the noise covariance is not sufficiently discussed. It’s unclear whether it’s crucial or not. For such a simple proposed idea, I’d expect much more rigorous ablation study for the key elements of the proposed method.
> >
> >
> >
> > Thanks for your comments. We will provide the sensitivity analysis of gaussian noise in the revision. More specially, we would like to analyze the effects of $\gamma$ for accuracy. The results are shown in the table:
> >
> >
> >
> > |Model |2e-3 | 1.5e-3 |1e-3 | 5e-4 |
> > |---------|------|---------|------|-------|
> > |WRN-28-10| 84.7 | 85.1 | 85.3 | 85.4 |
> > |ViT-B-16 | 79.7 | 80.2 | 80.8 | 80.8 |
> > |ViT-S-16 | 77.3 | 77.9 | 78.5 | 78.2 |
> >
> >
> >
> > From the above table, we can find that the noise plays an important role in the performance gain of R-SAM. There's a tradeoff but in general it's not hard to choose a good $\gamma$.
> >
> > ### **Improvement 6:**
> >
> > > The original SAM paper already pointed out that multiple steps of projected gradient ascent aren’t helpful (see Table 11 in https://arxiv.org/abs/2010.01412). That would be a more direct way to more accurately solve the inner maximization.
> >
> >
> >
> > Thanks for your  comments. This is a great question to further explore. We guess that multi-step also can obtain a better solution for the inner maximization problem, but the solution is different from the random smoothing based solution. For example, it is possible that there's a particular sharp region (or direction) where loss increases quickly; and minimizing this worst-case loss may not benefit too much to model's generalization. Instead, our method won't find those points due to the introduction of randomness. However, this is just our educational guess based on the experimental results, and we will further explore it in our future work.
> >
> > ### **Q1:**
> >
> >
> >
> > > 'However, the model usually locates in the sharp minima [31] where the unstable gradient, to a large extent, makes one-step gradient ascent performs poorly.' – This statement is surprising given that SAM aims to minimize sharpness and converge to a flat region. Moreover, this sentence leads to the impression that [31] demonstrated that this is somehow the case, although [31] has nothing to do with this.
> >
> >
> >
> > Thanks for your comments. We have removed this statement in the paper. We were intended to say that the inner maximization may not be well approximated with one-step gradient ascent in SAM. (and then you can talk about [31])
> >
> > Although [31] doesn't demonstrate the relationship between SAM and flatness, we still find that the loss value will be increased when we further perturb the weights of SAM. Based on that, we think SAM can converge to a more flat region than a traditional first-order optimizer, but still not very flat.
> >
> > ### **Q2:**
> >
> > > "Therefore, how to improve the approximation of the inner maximization procedure and obtain a more aggressive weight is an important problem” – What is meant by the “aggressive weight"?
> >
> >
> >
> > Sorry for the confusion. We have rewritten this sentence in the revision.
> >
> > ### **Q3:**
> >
> > > "However, the loss landscape of neural network is usually sharp and non-linear [30, 34]." – Surprising to see statements like this as the loss landscape of a linear model is also non-linear and may be considered sharp depending on the data covariance.
> >
> >
> > Sorry for the confusion. We have rewritten this statement in the revision.
> >
> > ### **Q4:**
> >
> > > "we observe that models can still locate in a sharp region, leading to poor performance on one-step gradient ascent." – This statement isn’t complemented by sharpness measurements.
> >
> >
> >
> > Sorry for the confusion. We have rewritten this part in the revision.
> >
> > ### **Q5:**
> >
> > > "we are able to apply the one-step gradient ascent on the smoothed weights for a much more accurate measurement of the inner maximization" –There is no evidence to this in the paper. Fig. 2 is done for a different algorithm.
> >
> >
> >
> > Thanks for your comments. As shown in the previous answers, Equation (6) and Equation (7) share the similar form, except with a difference scaling constant. Equation (6) can be represented as Equation (7) when we try to tune the value of $\delta$ and $\rho$ to make sure $\rho_{2} = \|\delta_{0}+\lambda g(\hat{w})\|$ and $\lambda = \frac{\rho_{1}}{\|g(w+\delta_{0})\|}$. If not, we can tune the value of $\rho_{2}$ and $\lambda$ to make sure that $\rho_{2} \frac{\lambda}{\|\delta_{0} + \lambda g(\hat{w})\|} = \frac{\rho_{1}}{\|g(w+\delta_{0})}$. In this way, the main difference between Equation (6) and Equation (7) will be the term about random noise $\delta_{0}$: $\delta_{0}$ in Equation (6) and $\frac{\rho_{2}}{\|\delta_{0} + \lambda g(\hat{w})\|}\delta_{0}$. Therefore, Equation (7) can be seen as introducing additional random noise compared with Equation (6).
> >
> > ### **Q6:**
> >
> > > Eq. (2): there should be a division by the gradient norm, not multiplication.
> >
> >
> >
> > Sorry for the confusion. We have corrected Equation (2) in the revision.

---

> > > ### Author Response · Authors · 2022-08-02
> > > **Response to Reviewer ciRx - Continue**
> > >
> > > ### **Q7:**
> > >
> > > > Was a grid search done for SAM (e.g., Table 1 and 2)? And how are the hyperparameters tuned for each dataset for the proposed method, R-SAM?
> > >
> > >
> > >
> > > Thanks for your comments. We use grid search for all the experimental results about SAM. The comparison between our reported results and the results in the paper of vanilla SAM are is shown in the Table:
> > >
> > >
> > >
> > > |Model |CIFAR-10 |CIFAR-100 |
> > > |--------------|----------|-----------|
> > > |Vanilla SAM | 97.3 | 83.5 |
> > > |Reported SAM | 97.3 | 84.6 |
> > >
> > >
> > >
> > > We can find that our reported result is the same as the result in vanilla SAM in CIFAR-10 and our reported accuracy is higher than vanilla SAM in CIFAR-100.

---

### Comment · Area_Chair_kUGo · 2022-08-07
**Discussion with Authors**

Dear Reviewers! Thank you so much for your time on this paper so far.

The authors have written a detailed response to your concerns. How does this change your review?

Please engage with the authors in the way that you would like reviewers to engage your submitted papers: critically and open to changing your mind. Thank you Reviewers tXqv and Dv1G for your initial engagement!

Looking forward to the discussion!

---

### Meta-Review · Area_Chair_kUGo · 2022-08-26

**Recommendation:** Accept
**Confidence:** Certain

**Metareview:**

All reviewers except one agreed that this paper should be accepted because of the strong author response during the rebuttal phase. Specifically the reviewers appreciated the significance of the problem being addressed, the clarity of the paper, the simplicity of the method, and the analysis. Authors: please carefully revise the manuscript based on the suggestions by the reviewers: they made many careful suggestions to improve the work and stressed that the paper should only be accepted once these changes are implemented. Once these are done the paper will be a nice addition to the conference!

**Award:**

No

---

### Decision · Program_Chairs · 2022-09-14

Accept